# *GovBench*: From Natural Language to Executable Pipelines, A New Benchmark for Data Governance Automation

## Abstract

Data governance is essential for scaling modern AI development. To automate data governance, numerous tools and models have emerged that translate user intent into executable governance code. However, the effectiveness of existing tools and models is largely unverified. The evaluation is severely hampered by the lack of a realistic, standardized, and quantifiable benchmark. This critical gap presents a significant obstacle to systematically evaluating utility and impedes further innovation in the field. To bridge this gap, we introduce Gov-Bench, a benchmark featuring a diverse set of tasks with targeted noise to simulate real-world scenarios and standardized scoring scripts for reproducible evaluation. Our analysis reveals that current data governance tools and models struggle with complex, multi-step workflows and lack robust error-correction mechanisms. We therefore propose DataGovAgent, a novel framework for end-to-end data governance utilizing a Planner-Executor-Evaluator architecture. This design incorporates contract-guided planning, retrieval from a reliable operator library, and sandboxed meta-cognitive debugging. Experimental results validate our approach: DataGovAgent significantly boosts the Average Task Score (ATS) on complex Directed Acyclic Graph (DAG) tasks from 39.7 to 54.9 and reduces debugging iterations by over 77.9% compared to general-purpose agent frameworks, a step toward more reliable automation of data governance. Code is available at https://anonymous.4open.science/r/GovBench-F6C6.

## 1 Introduction

Data fuels analytics and machine intelligence, yet the work required to make data trustworthy remains stubbornly manual. Studies report (Ahmadi et al., 2024) that practitioners spend the majority of their time cleaning, standardizing, integrating, and validating data rather than modeling it, turning skilled analysts into "data janitors" and creating a persistent bottleneck in the data value chain (Hosseinzadeh et al., 2023). Code-centric Extract, Transform, Load (ETL) pipelines and handwritten SQL/Python are powerful but brittle in the face of schema drift and data heterogeneity (Yang et al., 2025; Dinesh & Devi, 2024), costly to maintain, and slow to adapt to evolving business rules.

Large language models (LLMs) promise an alternative: specify governance intent in natural language and synthesize the required transformations automatically (Pahune & Chandrasekharan, 2025; Park et al., 2025a). However, progress is critically hampered by a significant evaluation gap. Existing benchmarks for automated data science often emphasize snippet-level coding or high-level analytics, failing to capture the unique challenges of data governance. They lack realistic, targeted noise, do not assess end-to-end workflows with business-grounded correctness, and cannot measure performance on complex, multi-step DAG pipelines.

To address this evaluation gap, we introduce GovBench, a hierarchically designed benchmark for natural-language-driven data governance. It contains 150 real-world tasks (100 operator-level; 50 DAG-level) covering six scenarios: Filtering, Refinement, Imputation, Deduplication & Consistency, Data Integration, and Classification & Labeling. GovBench's key innovations include: 1) a novel *"reversed-objective"* methodology—that inverts the original task goal to programmatically generate task-specific noise—to synthesize realistic and measurable noise; 2) a

longest-common-subsequence–aware (LCS-aware) sequencing algorithm that constructs compositionally deep DAG tasks with minimal pairwise overlap; and 3) auto-generated, task-specific evaluation scripts that provide normalized scores and standardized metrics—Code Runnable Rate (CRR), Task Success Rate (TSR), and Average Task Score (ATS)—ensuring a principled and reproducible assessment.

However, a robust benchmark is only half of the solution. When evaluated on GovBench, we find that even SOTA single-model (OpenAI, 2025; DeepSeek-AI & other authors, 2024; Hurst & other authors, 2024) baselines and general-purpose agent frameworks (Qian et al., 2024; Li et al., 2023) exhibit a significant performance gap. They struggle to decompose complex instructions, generate logically correct multi-step pipelines, and recover from errors, resulting in low task success rates. This reveals their architectural limitations: a lack of robust planning, insufficient grounding in reliable practices, and the absence of effective, structured debugging mechanisms.

To bridge this performance gap, we propose DataGovAgent, an end-to-end natural language to governance DAG (NL2GovDAG) framework specifically designed for the complexities of data governance. It translates natural language into verified governance DAGs through an Agentic Assembly Line of three specialized roles (Xi et al., 2025; Park et al., 2025b). Its core strengths are: 1) a Planner that employs contract-guided planning to ground user intent and propose a high-level DAG of abstract operators with machine-checkable guarantees; 2) an Executor that uses retrieval-augmented generation over a curated library (DCAI, 2025) of governance tools to reduce hallucination and improve code quality; and 3) an Evaluator that drives a meta-cognitive debugging loop in a sandbox, using contract violations to generate structured feedback until the code is both runnable and functionally correct.

On GovBench-150, DataGovAgent materially improves over strong single-turn baselines and competitive agent frameworks. With GPT-5 (OpenAI, 2025), it raises TSR from 49 to 64 on operator-level tasks (+15 pp) and from 46 to 60 on DAG-level tasks (+14 pp). Compared to the strongest agent baselines, ChatDev (Qian et al., 2024) , it lifts operator-level TSR from 43 to 64 (+21 pp) and, on DAG-level tasks, attains higher ATS (54.91 vs. 39.67, +15.24 points) and higher average score (mean of ATS, TSR, and CRR, 62.97 vs. 61.89, +1.08 points) while requiring 11.60 fewer debug iterations (Average Debug Iterations (ADI) 3.29 vs. 14.89).

In summary, our contributions are twofold:

- We introduce GovBench, the first hierarchical benchmark for data governance automation, which features 150 realistic tasks based on real-world sources, injected noise and a rigorous, multi-metric evaluation protocol to address the critical gap in assessing end-to-end pipeline correctness.
- We propose DataGovAgent, that significantly improves task success by translating natural language into verified governance pipelines through a unique combination of contract-guided planning, retrieval-augmented code generation, and meta-cognitive debugging.

## 2 RELATED WORK

### 2.1 DATA SCIENCE BENCHMARKS AND LLM EVALUATION

The rapid evolution of LLMs has catalyzed comprehensive evaluation frameworks for automated data science capabilities. Early benchmarks like DS-1000 (Lai et al., 2023) focused on snippet-level code generation for data science libraries, extended by DA-Code (Huang et al., 2024) for task-level evaluation in interactive environments. Recently, DataSciBench (Zhang et al., 2025), which provides systematic LLM agent evaluation with 25 multidimensional metrics across complete data science workflows, and ScienceAgentBench (Chen et al., 2025b), which targets rigorous assessment for data-driven scientific discovery, have been proposed (see Appendix A.1 for detailed benchmark comparison).

Contemporary evaluation has shifted toward sophisticated multidimensional assessment. HumanEval Pro (Yu et al., 2025) introduces self-invoking code generation requiring progressive reasoning capabilities, while mHumanEval (Raihan et al., 2025) extends multilingual code evaluation. LiveBench (White et al., 2025) addresses contamination issues in LLM evaluation with challenging,

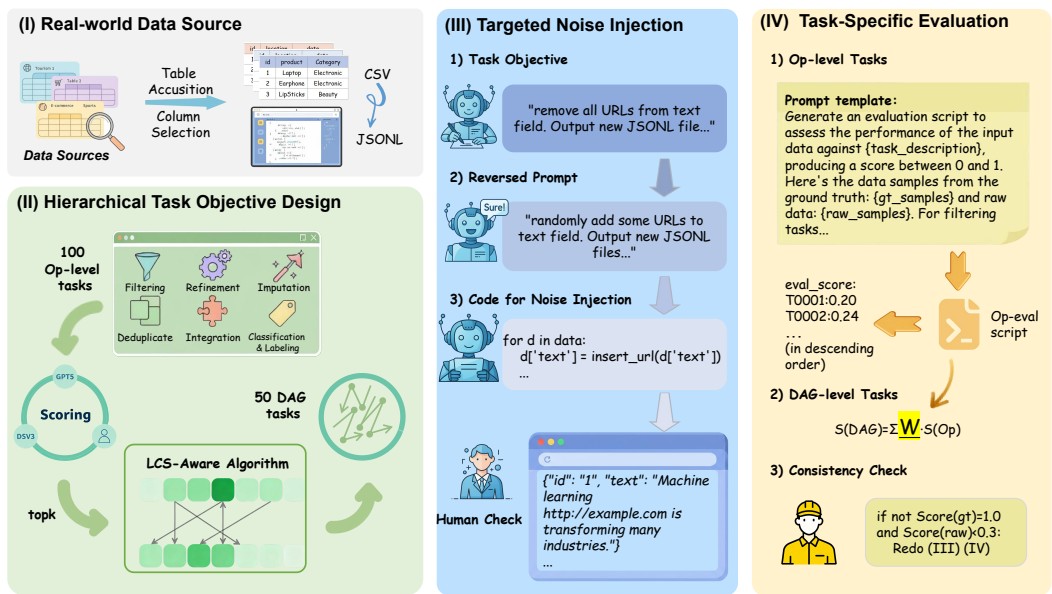

Figure 1: Illustration of the semi-automated pipeline designed for building GovBench, including real-world source data curation, hierarchical task objective design, targeted noise injection, and task-specific evaluation.

dynamic benchmarks (see Appendix A). These frameworks demonstrate significant performance variations, with SOTA models achieving 96.2% on HumanEval but declining to 76.2% on complex tasks.

## 2.2 DATA SCIENCE AGENTS AND AUTOMATION

Data science agents have evolved from simple code generators to comprehensive autonomous systems. Data Interpreter (Hong et al., 2025) employs hierarchical graph modeling for dynamic problem decomposition, while recent developments include AutoMind (Ou et al., 2025), offering adaptive knowledgeable agents for automated data science, and AutoML-Agent (Trirat et al., 2025), providing multi-agent frameworks for full-pipeline AutoML.

Current research emphasizes end-to-end workflow automation with minimal human intervention (Sun et al., 2024). TheAgentCompany (Xu et al., 2025) benchmarks LLM agents on consequential real-world tasks, while comprehensive surveys (Baek et al., 2025; Wang et al., 2024) highlight the transition from automation to autonomy in scientific discovery. These systems integrate planning, reasoning, reflection, and multi-agent collaboration capabilities. However, specialized data governance benchmarks remain limited. This gap highlights the necessity for benchmarks like our proposed **GovBench**.

Our work contributes through efficient data annotation pipelines generating customized evaluation scripts with standardized metrics including Code Runnable Rate (CRR), Task Success Rate (TSR), and Average Task Score (ATS), addressing gaps in governance-focused evaluation methodologies.

## 3  *GovBench:* A NEW BENCHMARK FOR DATA GOVERNANCE AUTOMATION

GovBench is a hierarchically designed data science benchmark dedicated to evaluating models' capabilities in performing data governance tasks. It comprises 150 real-world data governance problems, including 100 operator-level tasks and 50 DAG-level tasks. For each carefully curated NL task description, we synthesized ground-truth data and noisy data, accompanied by customized evaluation scripts to ensure precise and normalized scoring. GovBench comprehensively covers common scenarios encountered in real-life data governance workflows, including filtering, refinement, imputation, deduplication & consistency, data integration, and classification & labeling.

**Overview of Benchmark Creation.** To construct a hierarchical and realistic evaluation set for LLM-based data governance agents, we design a semi-automated pipeline comprising four stages: (1) data collection and column selection, (2) task objective definition and DAG construction via an LCS-aware algorithm, (3) noise injection, and (4) generation of task-specific evaluation scripts (see Figure 1; details in Sections 3.1–3.4). Statistics and examples are illustrated in Figure 4.

## 3.1 REAL-WORLD DATA SOURCE

To ensure comprehensive coverage of real-world scenarios, we curated 30 tables sourced from (Statista, 2025), spanning diverse domains such as tourism, eco-commerce, sports, and others. We retained only task-relevant columns (e.g., the `date` field for format normalization tasks) and necessary confounding columns (such as `birth_date`, which agents are not required to modify), thus maintaining data integrity and minimizing extraneous noise. Furthermore, to enhance problem diversity and facilitate flexible processing, the original CSV files were converted into JSONL format. These carefully selected and preprocessed datasets serve as the basis for synthesizing task descriptions, as detailed in Section 3.4.

## 3.2 HIERARCHICAL TASK OBJECTIVE DESIGN

GovBench comprises 100 Operator-level tasks and 50 DAG-level tasks. For Operator-level tasks, we designed six scenarios commonly encountered in real-world data governance, including filtering, refinement, imputation, deduplication &consistency, data integration, and classification/labeling. All tasks were carefully crafted by experienced data scientists to ensure clarity and fluency in their descriptions. The distribution of tasks in these scenarios is illustrated in **Figure 4**. For DAG-level tasks, we first rank the operator-level tasks by averaging the scores of GPT-5, DeepSeek-V3 (DeepSeek-AI & other authors, 2024), and the human baseline, thereby mitigating the bias introduced by relying solely on a single closed-source model, an open-source model, or human subjectivity. We then select 50 worst-performing Operator-level tasks as seed cases while treating the remaining tasks as candidates. We then introduce a simple yet efficient LCS-aware algorithm (see A.2) that takes existing tasks as input and generates task sequences. These sequences will be used to derive new DAG-level tasks objectives. This algorithm extends the required chain of thought while ensuring the complexity and diversity of DAG-level tasks by constraining different DAG tasks to share as few common sub-paths as possible, thereby presenting a substantial challenge to the model's capacity to handle intricate data governance problems. Given these sequences, we employ the prompt template provided in the **Prompt 1** to construct new natural language task objectives.

## 3.3 TARGETED NOISE INJECTION

The process of introducing noise into the dataset is divided into two distinct steps (Zhang et al., 2023; Akbiyik, 2023; Sousa et al., 2024). This method allows us to generate noisy data that will serve as a robust test set for evaluating the model's performance under imperfect conditions.

**Generate a Reversed Task Objective.** The first step involves generating a **reversed task objective** based on the provided data examples and the original task objective. This reversed objective shifts the focus from achieving the task goal (e.g., classification, imputation) to deliberately introducing noise into the data. For example, if the original task involves classifying data, the reversed task objective will focus on how to introduce noise such as mislabeling or irrelevant features. See the prompt template in **Prompt 2**.

**Generate Code to Introduce Noise.** In the second step, the model uses the reversed task objective, along with the provided data examples, to generate the actual code that will introduce the noise into the data. This code will implement the instructions described in the reversed objective—whether that involves adding missing values, creating duplicates, or generating irrelevant features. The goal is to transform the data in a way that makes it imperfect, allowing the model to be tested against noisy inputs. See the prompt template in **Prompt 3**.

At last, we manually check every data file, ensuring no extra noise is introduced because of model hallucination. This two-step approach allows for a targeted and methodical introduction of noise, ensuring that the noise is task-specific and realistic, which helps in robustly evaluating the model's performance.

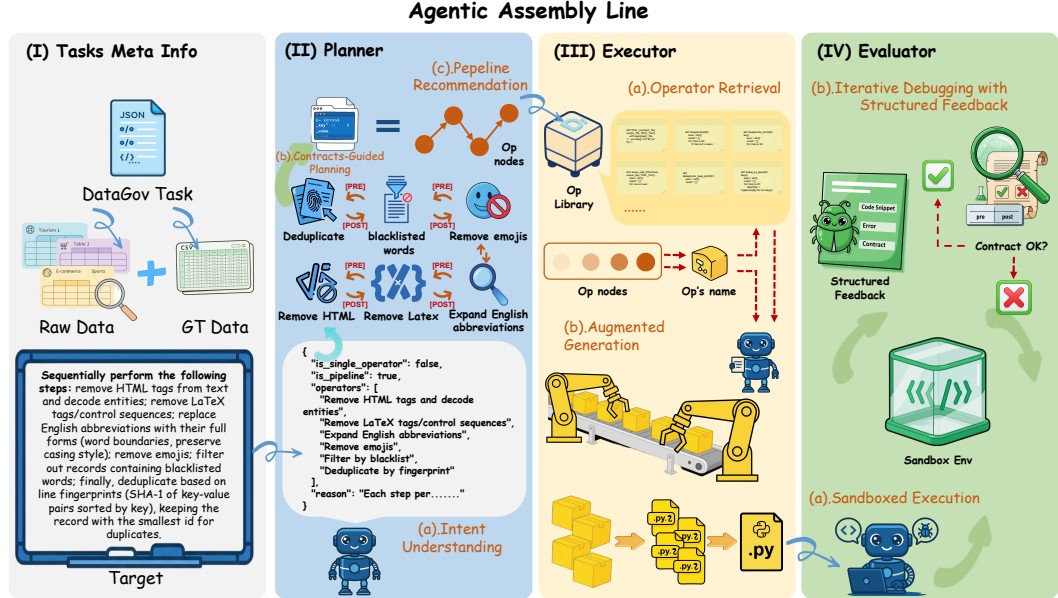

Figure 2: An overview of the Agentic Assembly Line, which progresses from intent understanding to contract-guided planning, followed by operator execution and sandboxed evaluation.

## 3.4 TASK-SPECIFIC EVALUATION

To evaluate the model's performance in handling noisy data, we design a prompt template to generate task-specific evaluation scripts. See the prompt template in **Prompt 4**. Each task's script compares the original dataset with the processed dataset and outputs a quantitative score between 0 and 1, reflecting the model's effectiveness in completing the task. Evaluation metrics are adjusted based on the specific nature of the task to ensure a precise assessment; a detailed breakdown for each Operator-level task category is provided in Table 7 in the **Appendix A.4**.

For DAG tasks, the final score is calculated based on the weighted average of scores from the operator-level tasks. We still use the average scores of GPT-5, DeepSeek-V3 (DeepSeek-AI & other authors, 2024), and the human baseline to calculate the weight, to mitigate the bias of any single source. The weights are determined by the following formula:

$$w_i = \frac{1}{1 + \alpha \cdot \text{score}_i} \tag{1}$$

Where $w_i$ is the weight of task $i$, $\alpha$ is a parameter that adjusts the influence of lower task scores, and $\text{score}_i$ is the average performance score of three solutions for each individual task.

**Consistency Check** After preparing the evaluation scripts, we run them on both the ground truth data and the input data. The ground truth should yield a score of 1.0, while the raw data should score below 0.3. If these conditions are not met, we manually adjust either the raw data or the scripts to ensure compliance with the standard.

## 4 *DataGovAgent:* AN END-TO-END NL2GOVDAG FRAMEWORK FOR DATA GOVERNANCE

To address the challenges of automating data governance, we introduce **DataGovAgent**, a novel multi-agent framework designed to interpret natural language instructions and autonomously orchestrate a DAG of data governance operations (Guo et al., 2024; Tran et al., 2025). The entire process, which we term **NL2GovDAG**, is operationalized through what we call an **Agentic Assembly Line**—a deterministic multi-agent workflow where specialized agents collaborate sequentially (Planner → Executor → Evaluator). Each step is governed by formal **governance contracts**, which are (pre, post) specifications that define input requirements and output guarantees for each operation. When execution fails, the system employs **meta-cognitive debugging**, an iterative refinement

process where agents reflect on their execution failures and generate targeted fixes based on contract violations and error analysis.

## 4.1 ARCHITECTURAL OVERVIEW

DataGovAgent employs an Agentic Assembly Line architecture (see Figure 2), enabling systematic decomposition and execution of data governance tasks through multi-agent collaboration.

## 4.2 SPECIALIZED AGENT ROLES

Our framework is instantiated by three core agent roles—the Planner, Executor, and Evaluator (Xu et al., 2024; Chen et al., 2025a). Their functions are orchestrated within a deterministic task chain, ensuring a structured progression from high-level intent to a verified, executable output.

Anchored in the data schema and representative samples, the ***Planner*** uses few-shot prompting to align user intent with the actual data and to assess feasibility; it then extracts machine-checkable **governance contracts** that formalize each operator as a (pre, post) tuple (Liu et al., 2024; Godboley & Krishna, 2025). Under these contracts, the Planner synthesizes an **initial DAG of abstract operators** such that the post-condition of each step satisfies the pre-condition of the next; when a constraint is not met, it inserts minimal repairs (for example, type casting or missing-value imputation) to ensure the pipeline is topologically coherent and executable.

For each DAG node, the ***Executor*** employs retrieval-augmented generation (Parvez et al., 2021; Trirat et al., 2025): it first retrieves the most relevant, validated operators from a curated library (DCAI, 2025) and then injects their descriptions and snippets as dynamic in-context exemplars to guide code synthesis, yielding Python implementations that are tailored to the task while adhering to established best practices, thereby reducing hallucinations and improving reuse.

The ***Evaluator*** executes the generated code in a restricted sandbox; upon any failure or noncompliance, it captures the offending code region, full error messages, and stack traces, and ties them to the violated contracts to produce targeted revision advice. This **meta-cognitive** feedback drives a guided correction loop until each operator is both runnable and contract-compliant, providing progressive validation on both construction and execution paths of the GovDAG. Implementation details and prompt templates are provided in **Appendix** A.6.

## 5 EXPERIMENTAL SETUP

To comprehensively evaluate the performance of DataGovAgent, we conducted systematic experiments on the newly constructed GovBench benchmark, covering experimental setup, evaluation metrics, baseline models, and results.

## 5.1 BENCHMARK

All experiments were conducted on the **GovBench-150** benchmark, which consists of 150 real-world data governance tasks designed to reflect the practical challenges faced by data scientists. Each single task provides a natural language description, the necessary raw dataset (s), and a custom evaluation script (`eval.py`) that objectively assesses output correctness with a normalized score in the range $[0, 1]$.

Tasks in GovBench-150 are categorized as either Operator-level—fine-grained tasks solvable with a single operation, such as filtering, format standardization, or simple imputation—or DAG-level tasks, which require coordinating multiple operations in a directed acyclic graph to accomplish complex, multi-step data cleaning, transformation, and integration.

## 5.2 EVALUATION METRICS

We employ the multi-dimensional metrics as shown in Table 10 to evaluate the performance of different models and frameworks.

Table 1: Performance of **Open-Source** Models on GovBench (Operator-Level)

| Model | ATS↑ | TSR↑ | CRR↑ | Avg. Score↑ | Avg. Tokens↓ | Generation Time (s)↓ | Execution Time (s)↓ |
|---|---|---|---|---|---|---|---|
| Qwen3-235b-a22b | 34.73 | 46.00 | 69.00 | 49.91 | 950.68 | 1,335.47 | 519.87 |
| Qwen2.5-coder | 27.99 | 38.00 | 58.00 | 41.33 | 589.57 | 1,039.39 | 81.26 |
| Qwen3-coder | 38.74 | 48.00 | 67.00 | 51.25 | 732.50 | 185.07 | 122.60 |
| DeepSeek-V3 | 35.68 | 47.00 | 74.00 | 52.23 | 680.51 | 1,663.45 | 572.13 |
| Llama-3-70B | 26.87 | 35.00 | 49.00 | 36.96 | 536.03 | 140.12 | 72.48 |
| Llama-4-scout | 14.88 | 23.00 | 37.00 | 24.96 | 702.50 | 618.06 | 151.65 |
| Mistral-7B | 10.41 | 15.00 | 27.00 | 17.47 | 715.78 | 525.99 | 87.74 |
| Gemma-3-27B | 29.62 | 43.00 | 76.00 | 49.54 | 1,425.84 | 4,042.13 | 60.92 |
| Phi4 | 23.24 | 32.00 | 42.00 | 32.41 | 982.37 | 1,642.61 | 98.73 |

Table 2: Performance of **Closed-Source** Models on GovBench (Operator-Level)

| Model | ATS↑ | TSR↑ | CRR↑ | Avg. Score↑ | Avg. Tokens↓ | Generation Time (s)↓ | Execution Time (s)↓ |
|---|---|---|---|---|---|---|---|
| GPT-5 | 40.98 | 49.00 | 81.00 | 56.99 | 3,706.21 | 3,069.44 | 598.73 |
| GPT-4o | 32.04 | 41.00 | 56.00 | 43.01 | 555.26 | 431.85 | 29.72 |
| o4-mini | 41.47 | 49.00 | 68.00 | 52.82 | 1,510.68 | 1,127.16 | 167.28 |
| o1 | 32.50 | 41.00 | 74.00 | 49.17 | 1,908.54 | 3,916.70 | 35.55 |
| o3 | 34.48 | 45.00 | 63.00 | 47.49 | 1,415.08 | 1,291.82 | 35.16 |
| Claude-4-sonnet | 36.75 | 46.00 | 85.00 | 55.92 | 1,672.91 | 3,149.83 | 229.70 |
| Claude-4-opus | 38.30 | 47.00 | 79.00 | 54.77 | 1,390.04 | 3,298.22 | 158.85 |
| Gemini-2.5-flash | 40.26 | 48.00 | 80.00 | 56.09 | 5,234.30 | 5,727.56 | 355.65 |
| Grok-3 | 35.41 | 44.00 | 71.00 | 50.14 | 688.51 | 811.22 | 685.25 |
| Grok-4 | 36.90 | 44.00 | 67.00 | 49.30 | 4,575.07 | 7,700.30 | 406.62 |
| Kimi-K2-instruct | 39.52 | 49.00 | 70.00 | 52.84 | 721.16 | 864.21 | 652.62 |

## 5.3 BASELINE

For a comprehensive comparison, we define three categories of baselines:

**Single-Model Baselines:** In this setting, the model receives the task description and must generate a complete solution in a single turn, without any multi-agent collaboration or self-debugging mechanisms. We evaluate mainstream open- and closed-source large language models.

**Agent Framework Baselines:** We select two representative multi-agent development frameworks—ChatDev (Qian et al., 2024) and CAMEL (Li et al., 2023)—adapt them to data-governance tasks, and use a strong closed-source model (e.g., GPT-5, GPT-4o) as the core engine to assess how existing agent frameworks perform on GovBench.

**Human Baseline:** We recruited five data science experts, each with over five years of experience in data engineering and analysis. To ensure a fair comparison, the experts were given unrestricted access to GPT-5 through a chat interface. They could ask any questions or request code suggestions as needed. However, they were required to manually synthesize, test, and refine the final Python solutions themselves.

## 6 BENCHMARK RESULTS

### 6.1 PERFORMANCE OF SINGLE-MODEL BASELINES

We evaluated the performance of single-model baselines on the operator-level tasks. The results are presented in Table 1 and Table 2.

From the performance of the single-model baselines, we observe the following:

**Significant Performance Ceiling:** Even the most powerful closed-source models, such as GPT-5 and Claude4-sonnet, fail to exceed a 50% TSR in a single-round code generation setting. This indicates that the tasks in GovBench are considerably challenging and difficult to solve perfectly with a single code generation attempt.

**Runnable Does Not Equal Correct:** Many models, such as **Claude4-sonnet**, exhibit a very high Code Runnability Rate (CRR > 80%), yet their TSR is significantly lower. This reveals a critical

Table 3: Performance of **Open-Source** Models on GovBench (DAG-Level)

| Model | ATS↑ | TSR↑ | CRR↑ | Avg. Score↑ | Avg. Tokens↓ | Generation Time (s)↓ | Execution Time (s)↓ |
|---|---|---|---|---|---|---|---|
| Qwen3-235b-a22b | 25.64 | 38.00 | 50.00 | 37.88 | 3,005.22 | 7,339.20 | 81.43 |
| Qwen2.5-coder | 12.11 | 26.00 | 30.00 | 22.70 | 738.68 | 852.36 | 28.23 |
| Qwen3-coder | 20.87 | 36.00 | 48.00 | 34.96 | 1,075.36 | 77.32 | 370.27 |
| DeepSeek-V3 | 28.65 | 56.00 | 72.00 | 52.22 | 983.70 | 1,098.90 | 305.99 |
| Llama-3-70B | 8.07 | 10.00 | 16.00 | 11.36 | 723.08 | 284.43 | 221.09 |
| Llama-4-scout | 7.35 | 12.00 | 22.00 | 13.78 | 864.16 | 435.08 | 10.39 |
| Mistral-7B | 7.10 | 18.00 | 20.00 | 15.03 | 897.88 | 261.90 | 230.13 |
| Gemma-3-27B | 11.31 | 20.00 | 38.00 | 23.10 | 1,671.34 | 2,412.24 | 19.06 |
| Phi-4 | 6.73 | 20.00 | 28.00 | 18.24 | 1,081.94 | 929.29 | 18.35 |

Table 4: Performance of **Closed-Source** Models on GovBench (DAG-Level)

| Model | ATS↑ | TSR↑ | CRR↑ | Avg. Score↑ | Avg. Tokens↓ | Generation Time (s)↓ | Execution Time (s)↓ |
|---|---|---|---|---|---|---|---|
| GPT-5 | 27.18 | 46.00 | 86.00 | 53.06 | 6,086.82 | 7,121.52 | 310.05 |
| GPT-4o | 18.68 | 38.00 | 50.00 | 35.56 | 754.82 | 276.54 | 52.94 |
| o4-mini | 31.86 | 56.00 | 74.00 | 53.95 | 2,075.26 | 971.14 | 91.31 |
| o1 | 27.79 | 52.00 | 80.00 | 53.26 | 2,574.00 | 3,270.06 | 15.68 |
| o3 | 31.22 | 46.00 | 64.00 | 47.07 | 2,027.76 | 1,410.07 | 85.07 |
| Claude-4-sonnet | 34.77 | 54.00 | 76.00 | 54.92 | 1,890.82 | 2,007.23 | 143.01 |
| Claude-4-opus | 20.41 | 34.00 | 50.00 | 34.80 | 1,759.84 | 2,443.04 | 74.24 |
| Gemini-2.5-flash | 25.40 | 44.00 | 68.00 | 45.80 | 7,383.40 | 2,457.91 | 295.21 |
| Grok-3 | 27.45 | 46.00 | 62.00 | 45.15 | 854.72 | 626.97 | 194.63 |
| Grok-4 | 31.38 | 50.00 | 66.00 | 49.13 | 5,537.42 | 4,706.45 | 277.36 |
| Kimi-K2-instruct | 20.60 | 30.00 | 34.00 | 28.20 | 1,107.94 | 758.61 | 80.78 |

issue: models can generate syntactically correct code, but the logic of this code does not necessarily meet the business objectives of the task.

**Potential of Open-Source Models:** Leading open-source code models, represented by **DeepSeek-V3**, can match or even surpass some closed-source models in TSR. This demonstrates their strong potential in the data science domain.

Building upon this, we have also systematically evaluated these models on the more challenging DAG-Level tasks. Unlike single-operator tasks, DAG tasks require the model to generate a **complete data processing workflow** in a single pass. This involves: 1) correctly decomposing the task into sub-tasks, 2) organizing them in a logical execution order, 3) ensuring correct dependency passing between steps, and 4) producing a final output that meets the specified business objectives. Due to the significant increase in complexity, the Avg. Score on DAG-Level tasks is generally lower than that on Operator-Level tasks.

Tables 3 and 4 summarize the baseline results for the open-source and closed-source models.

**Top-Tier Open-Source Models Rival Closed-Source Counterparts:** On DAG tasks, the leading open-source model, DeepSeek-V3 (DeepSeek-AI & other authors, 2024), achieved a 56.00 Task TSR. This performance not only leads the open-source field but also matches the top-performing closed-source model, o4-mini (56.00 TSR), while outperforming other powerful models like GPT-5 (46.00). This strongly indicates that leading open-source code models are highly competitive for handling complex, end-to-end data science workflows.

**Performance Divergence Among Closed-Source Models:** Within the closed-source camp, models exhibit different strengths. o4-mini demonstrates superior task-solving ability with the highest TSR. In contrast, Claude4-sonnet excels in ATS and Average Score, suggesting its generated code has higher overall quality and completeness. This reflects different optimization priorities among proprietary models.

The **"Runnable $\neq$ Correct"** Gap Is More Pronounced: In complex DAG tasks, the disparity between a high CRR and a low TSR is even more significant (e.g., GPT-5). For instance, GPT-5 shows an 86 CRR but only a 46 TSR. This reaffirms that generating syntactically correct complex workflows does not guarantee logical adherence to business objectives. Notably, the top-performing DeepSeek-V3 has a smaller gap between its CRR (72) and TSR (56), potentially indicating a better alignment between its code's runnability and its logical correctness.

**A Clear Trade-off Between Efficiency and Performance Persists:** The GPT-4o model demonstrates high generation efficiency, with the lowest token count and generation time among closed-source models. However, its 38.00 TSR is considerably lower than that of top-tier models. This highlights a clear trade-off between speed and accuracy when handling complex tasks, where some models achieve higher accuracy at a greater computational cost, while others are optimized for a balance between efficiency and performance.

## 6.2 PERFORMANCE OF AGENT FRAMEWORK BASELINES

We evaluated the ChatDev and CAMEL frameworks on GovBench by pairing them with powerful GPT-4o and GPT-5 models in Table 5.

Table 5: Performance of Agent Framework Baselines

(a) DAG-Level

| Framework | Base | ATS↑ | TSR↑ | CRR↑ | Avg. Score↑ | ADI↓ | Avg. Tokens↓ |
|---|---|---|---|---|---|---|---|
| ChatDev (Qian et al., 2024) | GPT-4o | 19.12 | 36.00 | 40.00 | 31.71 | 14.42 | 7,261.49 |
| ChatDev (Qian et al., 2024) | GPT-5 | 39.67 | 64.00 | 82.00 | 61.89 | 14.89 | 28,607.22 |
| CAMEL (Li et al., 2023) | GPT-4o | 8.47 | 24.00 | 60.00 | 30.82 | 5.00 | 11,925.00 |
| CAMEL (Li et al., 2023) | GPT-5 | 16.80 | 32.00 | 74.00 | 40.93 | 5.00 | 11,777.50 |
| DataGovAgent | GPT-4o | 34.52 | 44.00 | 50.00 | 42.84 | 4.03 | 27,192.45 |
| DataGovAgent | GPT-5 | 54.91 | 60.00 | 74.00 | 62.97 | 3.29 | 34,303.72 |

(b) Op-Level

| Framework | Base | ATS↑ | TSR↑ | CRR↑ | Avg. Score↑ | ADI↓ | Avg. Tokens↓ |
|---|---|---|---|---|---|---|---|
| ChatDev (Qian et al., 2024) | GPT-4o | 34.47 | 43.00 | 63.00 | 46.82 | 14.20 | 6,996.62 |
| ChatDev (Qian et al., 2024) | GPT-5 | 33.82 | 43.00 | 69.00 | 48.61 | 14.47 | 26,888.26 |
| CAMEL (Li et al., 2023) | GPT-4o | 14.54 | 29.00 | 91.00 | 44.85 | 4.40 | 9,071.92 |
| CAMEL (Li et al., 2023) | GPT-5 | 20.36 | 34.00 | 92.00 | 48.79 | 4.50 | 9,447.75 |
| DataGovAgent | GPT-4o | 52.93 | 63.00 | 89.00 | 68.31 | 2.12 | 23,712.14 |
| DataGovAgent | GPT-5 | 55.47 | 64.00 | 88.00 | 69.15 | 2.14 | 31,503.75 |

**Closing the Runnable–Correct Gap with Contracts and Meta-Cognitive Feedback:** On GovBench, DataGov-Agent consistently turns runnability into business-correct solutions more efficiently than generic agent frameworks. On DAG-level tasks, although ChatDev+GPT-5 attains the top TSR (64), DataGov-Agent+GPT-5 delivers higher average quality (ATS 54.91 vs. 39.67; Avg. Score 62.97 vs. 61.89), requires 4.5× fewer debug iterations (ADI 3.29 vs. 14.89). On operator-level tasks, DataGov-Agent+GPT-5 leads in TSR/ATS/Avg. Score (64/55.47/69.15) and shows the strongest alignment between runnability and correctness (A=TSR/CRR=0.73 vs. 0.62 for ChatDev and 0.37 for CAMEL), indicating that contracts and meta-cognitive feedback effectively convert CRR into TSR. More detailed analysis in **Appendix** A.7.

## 6.3 COMPARISON WITH HUMAN BASELINE

To contextualize the performance of DATAGOVAGENT, we conducted a comparative study against a strong human baseline of experienced data scientists, who were also aided by GPT-5. Our findings show a consistent pattern: on complex, multi-step DAG tasks, DATAGOVAGENT achieves higher accuracy and lower latency than the human baseline (TSR 60 vs. 25; 4.7 min vs. 24.5 min), whereas on operator-level tasks it is faster (3.5 min vs. 14.2 min) but less accurate (TSR 64 vs. 84). an in-depth discussion of the implications are provided in **Appendix** A.9.

## 7 CONCLUSION

We present GovBench, the first benchmark designed to comprehensively stress-test large language model agents on real-world data governance tasks. GovBench offers two main contributions: it provides a two-tiered task suite that spans from atomic operators to multi-step DAG pipelines, and for each task, it incorporates unique evaluation logic and scoring metrics. Furthermore, our proposed DataGovAgent achieves SOTA performance on this new benchmark, significantly outperforming existing agent frameworks on complex governance pipelines.

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

# A APPENDIX

## A.1 BENCHMARK COMPARISON TABLE

Table 6: Evolution of Code & Agent Benchmarks (textual overview; corresponding visual examples are shown in Figure 3).

| Benchmark | Evaluation Scope | Key Features | Methodological Focus |
|---|---|---|---|
| DS-1000 | Snippet-level | Code generation for data-science libraries (NumPy, Pandas) | Basic code completion |
| DA-Code | Task-level | Extends DS-1000 with an *interactive* execution environment | Interactive problem solving |
| DataSciBench | Workflow-level | Systematic LLM-agent evaluation with 25 multi-dimensional metrics | Complete data-science pipelines |
| ScienceAgentBench | Domain-specific | Rigorous assessment for data-driven scientific discovery | Scientific research workflows |
| HumanEval Pro | Reasoning-focused | Self-invoking code generation with progressive reasoning | Advanced reasoning capabilities |
| LiveBench | Methodology-focused | Dynamic benchmark that mitigates dataset contamination | Evaluation robustness |
| GovBench | Hierarchical (Operator & DAG-level) | 150 realistic tasks; reversed-objective noise; multi-metric scoring (ATS/TSR/CRR) | End-to-end data-governance pipeline evaluation |

```
{
  "prompt": "Given a numpy array a, compute the
mean value. result = ... # put solution in this
variable",
  "code_context": "...Python code for
test_execution and reference solution...",
  "metadata": {
    "lib": "Numpy",
    "difficulty": "Easy",
    ...
  }
}
```

(a) DS-1000

```
{
  "name": "titanic_survival_analysis",
  "description": "Given titanic.csv, perform
survival rate analysis grouped by gender.",
  "files": [
    {
      "filename": "titanic.csv",
      "filetype": "csv",
      "description": "Passenger data for Titanic
disaster"
    }
  ],
  "instruction": "Load titanic.csv and compute
......
```

(b) DA-Code

```
{
  "task_id": "1",
  "dependent_task_ids": [],
  "instruction": "Fine-tune the sentiment
classification model using the
EleutherAI/twitter-sentiment dataset",
  "task_type": "predictive modeling",
  "code": "tokenizer =
GPT2Tokenizer.from_pretrained('../gpt2-
small/')",
  "result": "",
  "is_success": true,
  "is_finished": true
},
```

(c) DataSciBench

```
{
  "id": "geoscience_01",
  "domain": "geoscience",
  "description": "Calculate the NDVI
(Normalized Difference Vegetation Index) for
a given area using satellite image data, and plot
the NDVI time series.",
  "inputs": [
    {
      "type": "raster",
      "name": "red_band",
      "description": "Red band image in GeoTIFF
format."
    ......
```

(d) ScienceAgentBench

```
{
  "task_id": "HumanEvalPro/1",
  "base_problem": {
    "prompt": "def add(a: int, b: int) -> int:\n
\"\"\"Add two integers.\"\"\"",
    "test_cases": [
      {"input": [1, 2], "output": 3},
      {"input": [-1, 5], "output": 4}
    ],
    "reference_solution": "def add(a: int, b: int) -
> int:\n    return a + b"
  },
  "pro_problem": { ......
```

(e) HumanEval Pro

```
{
  "question_id":
"0daa7ca38beec4441b9d5c04d0b98912322926
f0a3ac28a5097889d4ed83506f",
  "category": "reasoning",
  "ground_truth": "no, yes, yes",
  "turns": [
    "In this question, assume each person either
always tells the truth or always lies. Tala is at
the movie theater. The person at the
restaurant says the person at the aquarium lies.
Ayaan is at the aquarium. Ryan is at the
botanical garden. The person at the .....}
```

(f) LiveBench

```
{
  "task_id": "T0004",
  "target_en": "Please provide me with an
operator to process JSONL data by performing
the following text cleaning steps in sequence: (1)
remove extra spaces; (2) remove records whose
'text' field contains links (such as strings
starting with 'http://', 'https://', or 'www.'); (3)
remove records whose 'text' field is not in
English; (4) identify and remove records whose
'text' field contains spelling or grammatical
errors. The output should be in JSONL format,
UTF-8 encoded, with the original......
```

(g) GovBench

Figure 3: BenchDemo visual examples laid out two-per-row (last row has one). Compare with the textual description in Table 6.

## A.2 ALGORITHM FOR DERIVE OP-LEVEL TASK SEQUENCES

**Algorithm 1** LCS-constrained sequence synthesis. We randomly sample candidate sequences over the member set $M$ (without repetition, length 3/4/5), adjust conflicts using candidates from $C$, and finally output 50 valid sequences.

**Require:**
1: Member set $M = \{m_1, m_2, ..., m_{50}\}$ {Core task IDs}
2: Candidate set $C = \{c_1, c_2, ..., c_{50}\}$ {Replaceable task IDs}
**Ensure:**
3: Adjusted set $S_{\text{adjusted}}$ of size 50, satisfying:
- $\forall s \in S_{\text{adjusted}}, |s| \in \{3, 4, 5\}$
- $\forall s_i, s_j \in S_{\text{adjusted}}, \text{LCS}(s_i, s_j) \leq 1$

4: **Step 1: Random sampling of candidate sequences**
5: $S \leftarrow \emptyset$
6: $Budget \leftarrow 200$ {sample budget before adjustment}
7: **while** $|S| < Budget$ **do**
8:    $len \leftarrow \text{RandomChoice}(\{3, 4, 5\})$
9:    $seq \leftarrow \text{RandomSampleDistinct}(M, len)$ {prefer covering different items}
10:    $S \leftarrow S \cup \{seq\}$
11: **end while**
12: **Step 2: Conflict adjustment**
13: $S_{\text{adjusted}} \leftarrow \emptyset$
14: **for** $i = 0$ **to** $|S| - 2$ **do**
15:    **for** $j = i + 1$ **to** $|S| - 1$ **do**
16:       **while** $\text{ComputeLCS}(S[i], S[j]) \geq 2$ **and** $C \neq \emptyset$ **do**
17:          $lcs \leftarrow \text{GetLCS}(S[i], S[j])$
18:          $target\_seq \leftarrow (|S[i]| \geq |S[j]|)?S[i] : S[j]$
19:          $replace\_pos \leftarrow \text{RandomSelect}(\text{FindOccurrences}(lcs, target\_seq))$
20:          $c \leftarrow \text{RandomSelect}(C)$
21:          $target\_seq[replace\_pos] \leftarrow c$
22:          $C \leftarrow C \setminus \{c\}$
23:          $S_{\text{adjusted}} \leftarrow S_{\text{adjusted}} \cup \{target\_seq\}$
24:       **end while**
25:    **end for**
26: **end for**
27: **Step 3: Final selection**
28: $S_{\text{final}} \sim \text{UniformSample}(S_{\text{adjusted}}, 50)$
29: **return** $S_{\text{final}}$

## A.3 BENCHMARK STATISTICS

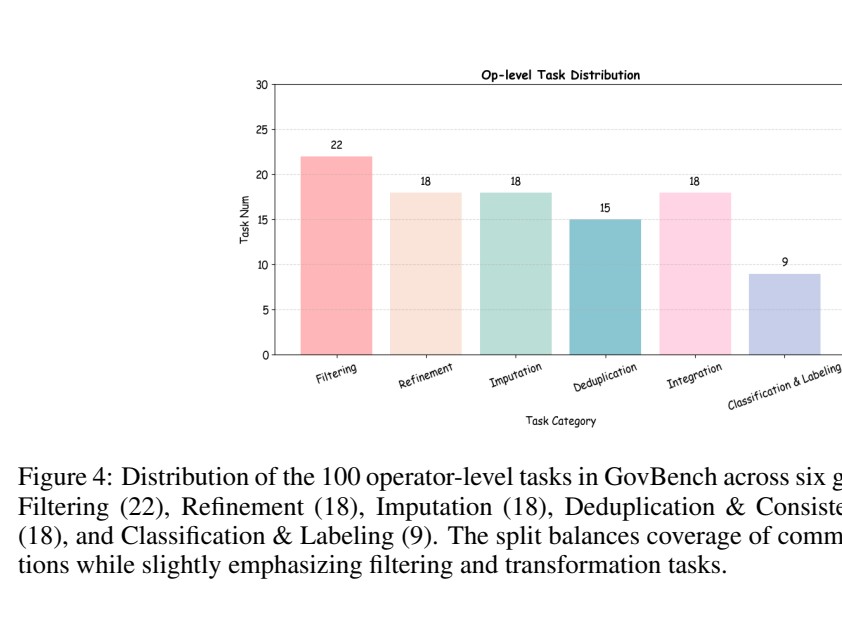

Figure 4: Distribution of the 100 operator-level tasks in GovBench across six governance categories: Filtering (22), Refinement (18), Imputation (18), Deduplication & Consistency (15), Integration (18), and Classification & Labeling (9). The split balances coverage of common governance operations while slightly emphasizing filtering and transformation tasks.

## A.4 TASKS EVAL

Table 7: Evaluation Metrics for Operator-Level Task Categories in GovBench

| Task Category | Primary Metric(s) | Description |
| --- | --- | --- |
| Filtering | F1 Score | Measures the balance of precision and recall in correctly identifying and removing erroneous or unwanted data rows. |
| Refinement | Accuracy | Assesses the correctness of data transformations, such as standardizing date formats, parsing text, or performing unit conversions. |
| Imputation | Completion Rate / Imputation Accuracy | Evaluates the model's effectiveness in correctly filling in missing or null values based on the ground truth. |
| Deduplication & Consistency | Duplicate Reduction / Consistency Score | Measures the success in identifying and removing duplicate records or ensuring that related data entries are consistent. |
| Data Integration | Integration Accuracy | Assesses how well data from different sources is merged, particularly in handling schema mismatches and resolving conflicting values. |
| Classification & Labeling | Accuracy, Precision, Recall, F1 Score | Uses standard classification metrics to evaluate the correctness of labels assigned to data records by the model. |

### A.5 BENCHMARK EXAMPLES

This part shows details of sample tasks across six Operator-level tasks and DAG tasks, including natural language task objectives, evaluation script snippets and dataset samples.

**1. Filtering Task**

---

**Filtering Task Objective**

Please write an operator to process jsonl files, filtering out text entries that contain blocked words (such as offensive, vulgar, or obscene words) in the text field. Each record is a JSON object, and it is necessary to check whether its text field contains blocked words. After filtering out these records, output a new JSONL file, keeping the field structure unchanged and encoded in UTF-8.

---

**Filtering Task Eval Code**

```python
def evaluate(expected_path, processed_path):
    expected = load_jsonl(expected_path)
    processed = load_jsonl(processed_path)

    expected_ids = set(entry['id'] for entry in expected)
    processed_ids = set(entry['id'] for entry in processed)

    true_positives = len(expected_ids & processed_ids)
    predicted_total = len(processed_ids)
    gold_total = len(expected_ids)

    precision = true_positives / predicted_total if predicted_total
     > 0 else 0.0
    recall = true_positives / gold_total if gold_total > 0 else 0.0
    f1 = (2 * precision * recall) / (precision + recall) if
    precision + recall > 0 else 0.0

    result = {"eval_score": f"{f1:.4f}"}
    print(result)
```

## Filtering Task Dataset Samples

| Raw Data | GT Data |
|---|---|
| { "id": 0, "text": "The customer service team is committed to providing exceptional assistance to all clients."} | { "id": 0, "text": "The customer service team is committed to providing exceptional assistance to all clients."} |
| { "id": 1, "text": "Customer service is dedicated to addressing and resolving client inquiries efficiently."} | { "id": 1, "text": "Customer service is dedicated to addressing and resolving client inquiries efficiently."} |
| { "id": 2, "text": "The refund policy stipulates that all requests must be submitted within thirty days of purchase."} | { "id": 2, "text": "The refund policy stipulates that all requests must be submitted within thirty days of purchase."} |
| { "id": 3, "text": "The refund policy allows customers to return products within 30 days for a full refund."} | { "id": 3, "text": "The refund policy allows customers to return products within 30 days for a full refund."} |
| { "id": 4, "text": "The product warranty is valid for a period of one year from the date of purchase."} | { "id": 4, "text": "The product warranty is valid for a period of one year from the date of purchase."} |
| "id": 32, "text": "Fix this crap service now." | |
| "id": 43, "text": "Fix this damn service now." | |
| "id": 58, "text": "This whole thing is total bitch." | |
| "id": 33, "text": "Why the bitch is my order delayed?" | |
| "id": 83, "text": "Why the bastard is my order delayed?" | |

## 2. Refinement Task

### Refinement Task Objective

Please write an operator to process JSONL files and remove HTML tags from the text field. Each record is a JSON object, requiring detection of its text field and removal of all HTML tags (e.g., <p>, <a href='url'>, etc.). Output a new JSONL file, retaining the field structure unchanged, encoded in UTF-8.

### Refinement Task Eval Code

```python
def evaluate(expected_path, processed_path, show_diff=5):
    expected = load_jsonl(expected_path)
    processed = load_jsonl(processed_path)

    total = len(expected)
    matched = 0
    mismatches = []

    for id_, exp_text in expected.items():
        proc_text = processed.get(id_)
        if proc_text is None:
            mismatches.append((id_, "missing", exp_text, ""))
        else:
            if normalize(proc_text) == normalize(exp_text):
                matched += 1
            else:
```

```
                mismatches.append((id_, "mismatch", exp_text,
    proc_text))

    accuracy = matched / total if total > 0 else 0.0

    result = {"eval_score": f"{accuracy:.4f}"}
    print(result)
```

## Refinement Task Dataset Samples

| Raw Data | GT Data |
|---|---|
| { "id": "id_0001", "topic": "climate change", "text": "Climate change poses significant challenges to the global environment and necessitates urgent collective action." } | { "id": "id_0001", "topic": "climate change", "text": "Climate change poses significant challenges to the global environment and necessitates urgent collective action." } |
| { "id": "id_0002", "topic": "climate change", "text": "Climate change poses a significant threat to the stability of ecosystems worldwide." } | { "id": "id_0002", "topic": "climate change", "text": "Climate change poses a significant threat to the stability of ecosystems worldwide." } |
| { "id": "id_0003", "topic": "climate change", "text": "Climate change poses a significant threat to global ecosystems and human societies." } | { "id": "id_0003", "topic": "climate change", "text": "Climate change poses a significant threat to global ecosystems and human societies." } |
| { "id": "id_0004", "topic": "climate change", "text": "Climate change poses a significant threat to global ecosystems and human societies." } | { "id": "id_0004", "topic": "climate change", "text": "Climate change poses a significant threat to global ecosystems and human societies." } |
| { "id": "id_0005", "topic": "climate change", "text": "Climate change presents a significant challenge that requires immediate global attention and action." } | { "id": "id_0005", "topic": "climate change", "text": "Climate change presents a significant challenge that requires immediate global attention and action." } |

## 3. Imputation Task

### Imputation Task Objective

Need a data governance operator that uses the KNN algorithm (k=3) to impute missing values in a CSV file. 1. Input file: CSV (with header, comma-separated). 2. Supports numeric and one-hot encoded categorical variables. Encoding: UTF-8, no BOM.

### Imputation Task Eval Code

```python
def evaluate(cand: pd.DataFrame,
             gt: pd.DataFrame,
             raw: pd.DataFrame) -> float:

    if cand.shape != gt.shape:
        fail(f"Mismatch in dimensions: Expected {gt.shape}, Actual {cand.shape}")
    if list(cand.columns) != list(gt.columns):
        fail("Column names or order do not match the reference")
```

```
    miss_mask = raw.isna()

    if cand[miss_mask].isna().any().any():
        fail("There are missing values that were not filled")

    diff = np.abs(cand[miss_mask].astype(float) - gt[miss_mask].
    astype(float))
    if (diff > ATOL).any().any():
        fail("The filled values do not match the reference (non-KNN
     imputation)")

    if not cand[~miss_mask].astype(float).equals(raw[~miss_mask].
    astype(float)):
        fail("The originally complete data has been modified")

    return 1.0
```

**Imputation Task Dataset Samples**

| Raw Data | GT Data |
|---|---|
| customer_id, age, income, color_blue, color_green, color_red | customer_id, age, income, color_blue, color_green, color_red |
| 1, 22.0, 37110.61305675143, True, False, False | 1, 22.0, 37110.61305675143, 1.0, 0.0, 0.0 |
| 2, 58.0, 55531.26176123748, False, False, True | 2, 58.0, 55531.26176123748, 0.0, 0.0, 1.0 |
| 3, 52.0, 35616.760987565016, False, False, True | 3, 52.0, 35616.760987565016, 0.0, 0.0, 1.0 |
| 4, 40.0, 63176.75451960909, True, , | 4, 40.0, 63176.75451960909, 1.0, 0.3333333333333333, 0.3333333333333333 |
| 5, 40.0, 49251.11133520621, False, True, False | 5, 40.0, 49251.11133520621, 0.0, 1.0, 0.0 |
| 6, 62.0, 47227.06454682109, False, , | 6, 62.0, 47227.06454682109, 0.0, 0.0, 0.3333333333333333 |
| 7, 22.0, 39786.05683394088, True, False, False | 7, 22.0, 39786.05683394088, 1.0, 0.0, 0.0 |
| 8, 54.0, 68338.12008011046, False, , True | 8, 54.0, 68338.12008011046, 0.0, 0.0, 1.0 |
| 9, 28.0, 47682.05776896797, True, False, False | 9, 28.0, 47682.05776896797, 1.0, 0.0, 0.0 |
| 10, 22.0, 43575.08266755339, False, False, True | 10, 22.0, 43575.08266755339, 0.0, 0.0, 1.0 |
| 11, 45.0, , True, False, | 11, 45.0, 58632.88840075844, 1.0, 0.0, 0.0 |
| 12, 68.0, 57984.63778330023, True, False, False | 12, 68.0, 57984.63778330023, 1.0, 0.0, 0.0 |
| 13, , 55481.660965461175, True, False, | 13, 54.333333333333336, 55481.660965461175, 1.0, 0.0, 0.3333333333333333 |
| 14, 57.0, 56190.98917393983, False, True, False | 14, 57.0, 56190.98917393983, 0.0, 1.0, 0.0 |
| 15, 55.0, 56462.315045118245, , True, False | 15, 55.0, 56462.315045118245, 0.6666666666666666, 1.0, 0.0 |

## 4. De-duplication Task

**De-duplication Task Objective**

A data governance operator for incremental deduplication on `*.csv`/`*.jsonl`: 1. Historical baseline: `.jsonl` (already deduplicated, contains id, updated_at, and business fields) 2. New incremental file: `.csv` (same structure) 3. Primary key: id 4. Deduplication rules: If the primary key exists in the baseline, ignore the incremental row; if not, append to the result set; For the same key but different business fields, keep the record with the latest updated_at.

## De-duplication Task Eval Code

```python
def compute_f1(
    gt_map: Dict[str, Dict],
    pred_rows: List[Dict],
) -> float:
    if not pred_rows:
        return 0.0

    tp_ids: Set[str] = set()
    fp = 0

    for row in pred_rows:
        rid = str(row.get("id", ""))
        if not rid:
            fp += 1
            continue

        # Duplicate row
        if rid in tp_ids:
            fp += 1
            continue

        gt_row = gt_map.get(rid)
        if gt_row is None:
            fp += 1  # Extra id
            continue

        # Compare all fields with GT (order doesn't matter)
        if row == gt_row:
            tp_ids.add(rid)
        else:
            fp += 1  # Field values do not match

    fn = len(gt_map) - len(tp_ids)
    precision = len(tp_ids) / (len(tp_ids) + fp) if tp_ids or fp
    else 0.0
    recall = len(tp_ids) / (len(tp_ids) + fn) if tp_ids or fn else
    0.0
    if precision + recall == 0:
        return 0.0
    return 2 * precision * recall / (precision + recall)
```

## De-duplication Task Dataset Samples

| Raw Data | GT Data |
|---|---|
| **File1:** { ”id”: ”C0061”, ”updated_at”: ”2025-04-20T13:59:30Z”, ”name”: ”Isaac”, ”tier”: ”gold” } | { ”id”: ”C0001”, ”updated_at”: ”2024-01-15T10:30:00Z”, ”name”: ”Alice”, ”tier”: ”gold” } |
| { ”id”: ”C0024”, ”updated_at”: ”2024-07-10T13:21:47Z”, ”name”: ”Xavier”, ”tier”: ”bronze” } | { ”id”: ”C0002”, ”updated_at”: ”2024-02-03T08:14:12Z”, ”name”: ”Bob”, ”tier”: ”silver” } |
| { ”id”: ”C0094”, ”updated_at”: ”2025-12-07T09:03:25Z”, ”name”: ”Queen”, ”tier”: ”gold” } | { ”id”: ”C0003”, ”updated_at”: ”2024-02-27T19:22:05Z”, ”name”: ”Carol”, ”tier”: ”bronze” } |
| { ”id”: ”C0094”, ”updated_at”: ”2025-12-07T09:03:25Z”, ”name”: ”Queen”, ”tier”: ”gold” } | { ”id”: ”C0004”, ”updated_at”: ”2024-03-10T07:45:51Z”, ”name”: ”Dave”, ”tier”: ”gold” } |
| { ”id”: ”C0075”, ”updated_at”: ”2025-07-27T08:12:05Z”, ”name”: ”Xander”, ”tier”: ”bronze” }... | { ”id”: ”C0005”, ”updated_at”: ”2024-03-19T11:26:31Z”, ”name”: ”Eve”, ”tier”: ”silver” } |
| **File2:** id,updated_at,name,tier | { ”id”: ”C0006”, ”updated_at”: ”2024-03-27T15:02:43Z”, ”name”: ”Frank”, ”tier”: ”bronze” } |
| C0068,2025-06-25T00:05:48Z,Paula,silver | { ”id”: ”C0007”, ”updated_at”: ”2024-04-02T09:56:17Z”, ”name”: ”Grace”, ”tier”: ”gold” } |
| C0107,2025-08-06T05:37:13Z,New107,silver | { ”id”: ”C0008”, ”updated_at”: ”2024-04-11T20:11:00Z”, ”name”: ”Heidi”, ”tier”: ”silver” } |
| C0072,2025-07-24T11:00:49Z,Una,gold | { ”id”: ”C0009”, ”updated_at”: ”2024-04-23T05:33:29Z”, ”name”: ”Ivan”, ”tier”: ”bronze” } |
| C0062,2025-05-27T05:43:16Z,Jane,silver | { ”id”: ”C0010”, ”updated_at”: ”2024-04-30T18:44:07Z”, ”name”: ”Judy”, ”tier”: ”gold” }... |
| C0018,2024-07-21T07:27:37Z,Rupert,gold... | |

## 5. Integration Task

### Integration Task Objective

A data governance operator for composite key join: join by multi-column composite keys and resolve column conflicts. Input: customer1.csv, customer2.csv. Rule: Composite key: left(k1,k2,...) = right(k1',k2',...) (same number of columns). Conflict resolution: left-priority/right-priority/left and right suffix. Output: gt.csv.

### Integration Task Eval Code

```python
def evaluate(gt_hdr: List[str],
             gt_rows: List[Dict[str, str]],
             pred_rows: List[Dict[str, str]]) -> float:
    # 1. Column completeness
    if not pred_rows:
        print("[eval] Output is empty", file=sys.stderr)
        return 0.0

    missing = [c for c in gt_hdr if c not in pred_rows[0]]
    if missing:
```

```
        print(f"[eval] Missing columns: {missing}", file=sys.stderr
    )
        return 0.0

    # 2. Set comparison
    gt_counter   = rows_to_counter(gt_rows,   gt_hdr)
    pred_counter = rows_to_counter(pred_rows, gt_hdr)

    if gt_counter != pred_counter:
        lack  = gt_counter - pred_counter
        extra = pred_counter - gt_counter
        if lack:
            print(f"[eval] Missing row examples: {list(lack.
    elements())[:3]} ...",  file=sys.stderr)
        if extra:
            print(f"[eval] Extra row examples: {list(extra.elements
    ())[:3]} ...", file=sys.stderr)
        return 0.0

    return 1.0
```

**Integration Task Dataset Samples**

| Raw Data | GT Data |
| --- | --- |
| **File1:**
country,region,customer_id,email, signup_date,status,notes
US,CA,1001,alice@example.com,2021-01-10,active,L1
US,NY,1002,bob@example.com,2021-02-12,inactive,L2
CN,BJ,2001,chen@example.cn,2020-11-05,active,L3
CN,SH,2002,du@example.cn,2022-07-19,pending,L4
DE,BE,3001,eva@example.de,2021-09-30,active,L5
US,CA,1003,frank@example.com,2020-06-15,active,L6
**File2:**
country_code,region,id,email, last_order_date,status,vip
US,CA,1001,alice.us@example.com, 2022-12-01,gold,true
US,NY,1002,bob@example.com,2021-12-11,inactive,false
CN,BJ,2001,chen_new@ex.cn,2023-03-03,active,true
CN,GD,2005,gao@example.cn,2021-05-05,active,false
DE,BE,3001,eva@example.de, 2022-02-02,paused,false
US,CA,9999,zoe@example.com,2023-04-04,active,false
US,CA,1003,frank@example.com,2020-07-01,inactive,false
CN,SH,2002,du@alt.cn,2022-08-01,active,true | country,region,customer_id,email_left, signup_date,status_left,notes,email_right last_order_date,status_right,vip
US,CA,1001,alice@example.com,2021-01-10,active,L1,alice.us@example.com, 2022-12-01,gold,true
US,NY,1002,bob@example.com,2021-02-12,inactive,L2,bob@example.com, 2021-12-11,inactive,false
CN,BJ,2001,chen@example.cn,2020-11-05,active,L3,chen_new@ex.cn,2023-03-03,active,true
CN,SH,2002,du@example.cn,2022-07-19,pending,L4,du@alt.cn,2022-08-01,active,true
DE,BE,3001,eva@example.de,2021-09-30,active,L5,eva@example.de,2022-02-02,paused,false
US,CA,1003,frank@example.com,2020-06-15,active,L6,frank@example.com, 2020-07-01,inactive,false |

## 6. Classification and Labeling Task

**Classification and Labeling Task Objective**

Use LLMserving to assign sentiment labels to text: Input format: `.jsonl` with `text_id` and `content`; Sentiment label set: Positive / Neutral / Negative.

**Classification and Labeling Task Eval Code**

```python
def accuracy(gt: List[Dict[str, Any]], pred: List[Dict[str, Any]])
    -> float:
    """
    Calculate the simple classification accuracy between
    predictions and ground truth.
    """
    # Create {text_id: sentiment} mapping; trim leading and
    trailing spaces and standardize case
    norm = lambda s: str(s).strip()  # Only trim; case-sensitive
```

```
    gt_map   = {norm(r["text_id"]): norm(r["sentiment"]) for r in
    gt}
    pred_map = {norm(r["text_id"]): norm(r.get("sentiment", ""))
    for r in pred}

    total   = len(gt_map)
    correct = sum(1 for k, v in gt_map.items() if pred_map.get(k)
    == v)
    return correct / total if total else 0.0
```

**Classification and Labeling Task Dataset Samples**

| Raw Data | GT Data |
|---|---|
| {"text_id": "0001", "content": "The latte at this coffee shop is so delicious, I will definitely come back next time!"}
{"text_id": "0002", "content": "The customer service response speed is quite fast, and the problem has been solved."}
{"text_id": "0003", "content": "The sunlight today is really nice, feeling great."}
{"text_id": "0004", "content": "The soundtrack of this movie is very moving, definitely recommend it."}
{"text_id": "0005", "content": "The project was launched on time, and everyone is very satisfied."}
{"text_id": "0079", "content": "This is the second page of the contract."}
{"text_id": "0080", "content": "The air conditioning temperature is set to 25°C."}
{"text_id": "0081", "content": "The service attitude was terrible, I will never come again."}
{"text_id": "0082", "content": "The product broke after just two days of use, very disappointing."}
{"text_id": "0083", "content": "The courier hasn't updated the logistics for a week, so annoying."} | {"text_id":"0001","content":"The latte at this coffee shop is so delicious, I will definitely come back next time!","sentiment":"Positive"}
{"text_id":"0002","content":"The customer service response speed is quite fast, and the problem has been solved.","sentiment":"Positive"}
{"text_id":"0003","content":"The sunlight today is really nice, feeling great.","sentiment":"Positive"}
{"text_id":"0004","content":"The soundtrack of this movie is very moving, definitely recommend it.","sentiment":"Positive"}
{"text_id":"0005","content":"The project was launched on time, and everyone is very satisfied.","sentiment":"Positive"}
{"text_id":"0079","content":"This is the second page of the contract.","sentiment":"Neutral"}
{"text_id":"0080","content":"The air conditioning temperature is set to 25°C.","sentiment":"Neutral"}
{"text_id":"0081","content":"The service attitude was terrible, I will never come again.","sentiment":"Negative"}
{"text_id":"0082","content":"The product broke after just two days of use, very disappointing.","sentiment":"Negative"}
{"text_id":"0083","content":"The courier hasn't updated the logistics for a week, so annoying.","sentiment":"Negative"} |

## 7. DAG Task

---
**DAG Task Objective**

Write an operator to process JSONL files, executing sequentially: filter out records with a high proportion of symbols in the text field → remove excess spaces in the text field → censor profanity in the text field with ****, for example, "I am fucking happy" becomes "I am **** happy" → use MinHash for approximate deduplication ($\geq$0.9), retaining the record with the smallest id; output JSONL.

---
**DAG Task Eval Code**

```python
def evaluate(processed_path):
    expected_path = get_gt()
    expected = load_jsonl(expected_path)
    processed = load_jsonl(processed_path)

    # Construct mappings for comparison
    expected_map = {entry["id"]: entry for entry in expected}
    processed_map = {entry["id"]: entry for entry in processed}

    # Only evaluate the intersection part
    common_ids = set(expected_map.keys()) & set(processed_map.keys
    ())

    true_positives = 0
    for cid in common_ids:
        gt = expected_map[cid]
        pred = processed_map[cid]

        # Check if text is the same (strip leading and trailing
    spaces)
        if gt["text"].strip() == pred["text"].strip():
            true_positives += 1

    predicted_total = len(processed_map)
    gold_total = len(expected_map)

    precision = true_positives / predicted_total if predicted_total
     > 0 else 0.0
    recall = true_positives / gold_total if gold_total > 0 else 0.0
    f1 = (2 * precision * recall) / (precision + recall) if
    precision + recall > 0 else 0.0

    result = {"eval_score": f"{f1:.4f}"}
    print(result)
```

---

**DAG Task Dataset Samples**

| Raw Data | GT Data |
|---|---|
| {"id": 1, "items": ["orange", "computer", "paper", "pear", "book", "phone"], "text": "Sports and the environment have a complex relationship that requires careful consideration and action if we want to keep enjoying both. On one hand, sporting events bring people together, promote health, and drive the economy. On the other hand, they can be a asshole environmental nightmare", "sources": ["dataset_b.jsonl", "dataset_a.jsonl"]} | {"id": 1, "items": ["orange", "computer", "paper", "pear", "book", "phone"], "text": "Sports and the environment have a complex relationship that requires careful consideration and action if we want to keep enjoying both. On one hand, sporting events bring people together, promote health, and drive the economy. On the other hand, they can be a **** environmental nightmare", "sources": ["dataset_b.jsonl", "dataset_a.jsonl"]} |
| {"id": 26, "items": ["orange", "book", "banana", "grape", "computer"], "text": "Engaging in sports is one hell of a way to boost your overall health and well-being, both physically and mentally. Whether you're hitting the gym, playing soccer, or going for a run, these activities keep your", "sources": ["dataset_b.jsonl", "dataset_c.jsonl"]} | {"id": 26, "items": ["orange", "book", "banana", "grape", "computer"], "text": "Engaging in sports is one hell of a way to boost your overall health and well-being, both physically and mentally. Whether you're hitting the gym, playing soccer, or going for a run, these activities keep your", "sources": ["dataset_b.jsonl", "dataset_c.jsonl"]} |
| {"id": "d4a6cae8-6250-40dc-9a1e-b9bef91620fd", "items": ["pen", "orange", "grape", "computer", "banana", "paper"], "text": "Art has a **** magical way of weaving itself into the fabric of health, providing both mental clarity and emotional solace. Through the **** strokes of a paintbrush ????????????????????????????????????? ????????????????????????????????????? ????????????????????????????????????? or the rhythmic beats of a song, art offers a therapeutic escape from life's", "sources": ["dataset_b.jsonl"]} | |

## A.6 AGENT ROLES AND IMPLEMENTATION DETAILS

**The Planner: From Intent to High-Level DAG.** The initial phase is dedicated to understanding the user's goal and formulating a strategic plan. This is achieved through two sequential tasks:

- *Intent Understanding:* Upon receiving a natural language request, the Planner leverages a LLM configured with **few-shot prompting**. It analyzes the user's intent by conditioning the model with the provided data schema and data samples. This grounding process ensures the user's goal is not only correctly interpreted but also validated for feasibility against the actual data context.

- *Contract-Guided Planning:* After intent understanding, the Planner does not directly generate a concrete blueprint. Instead, it first extracts machine-checkable governance contracts from the user request, data schema, and data samples. Each contract is attached to an operator in the form of a 2-tuple (PRE, POST), strictly defining the pre-conditions and post-conditions for execution. The Planner then generates a sequence that satisfies the constraints imposed by these contracts, ensuring that the output (POST) of each step fulfills the input requirements (PRE) of the subsequent step. When a constraint is not met, the system automatically inserts minimal repair steps (such as imputation or type casting).

- *Pipeline Recommendation:* Building on the above deep understanding and contract-guided planning, the Planner ultimately formulates a high-level governance plan, which is represented as a preliminary directed acyclic graph (DAG). The nodes of this DAG correspond to a series of abstract operators (e.g., "Remove Duplicates", "Standardize Date Format", "Impute Missing Values"). These contract-annotated nodes collectively provide a strategic blueprint for the subsequent execution phase, ensuring that the final generated code strictly adheres to the validated logical path.

**The Executor: Realizing Operators with Retrieval-Augmented Generation.** For each abstract operator in the planned DAG, the Executor is responsible for generating concrete, executable Python code. It employs a powerful **Retrieval-Augmented Generation (RAG)** strategy, which synergizes the reliability of pre-validated code with the flexibility of on-the-fly generation.

- *Operator Retrieval:* The agent first treats its internal library of validated governance operators as a collection of callable **tools**. Each tool has a rich description detailing its functionality, parameters, and use cases. The Executor compares the semantic content of the target operator's goal (e.g., "standardize date format to YYYY-MM-DD") against these tool descriptions to retrieve the top-K (e.g., top-4) most relevant operators.

- *Augmented Generation:* Rather than simply executing the top retrieved operator or falling back to free generation if no perfect match is found, the Executor adopts a more robust approach. The retrieved operators, along with their descriptions, are injected into the LLM's prompt as dynamic few-shot examples. This context-rich prompt guides the model to generate code that is not only tailored to the specific requirements of the task but also adheres to the established patterns and best practices of the operator library. This hybrid method significantly reduces hallucinations and improves the quality of the generated code, even for highly customized or novel tasks.

**The Evaluator: Sandboxed Execution and Meta-Cognitive Refinement.** Code generation is only half the battle; rigorous verification is paramount. The Evaluator provides a critical quality assurance layer through a self-correcting execution and debugging cycle.

- *Sandboxed Execution:* All generated code is executed within a secure, isolated sandbox environment. This prevents unintended side effects on the host system and allows the agent to safely handle diverse data sources and external dependencies.

- *Iterative Debugging with Structured Feedback:* When the generated code fails to execute or produces incorrect results, the Evaluator does not simply report the failure. Instead, it acts as a diagnostician, capturing the runtime state and constructing a highly structured feedback prompt to guide the Executor's subsequent refinement. As shown in Figure 2, this prompt is a rich data object containing a comprehensive diagnostic report: it includes not only the erroneous code snippet that caused the failure, but also the complete error message and stack trace, providing technical context for issue localization. More importantly, the Evaluator also analyzes the situation in light of the relevant **contract constraints**. If any contract is found to be unsatisfied, it offers targeted revision suggestions—for example, *Please add a check to handle potential null values in the creation_date column before applying the datetime conversion."* To keep the agent aligned with the overall objective, the feedback additionally includes broader task context.

This *meta-cognitive feedback* allows the Executor to perform targeted, surgical corrections instead of trial-and-error guessing. This loop continues until the operator code is both runnable and functionally correct, ensuring each component of the final GovDAG is rigorously validated.

### A.7 DETAILS OF AGENT FRAMEWORK BASELINES

#### A.7.1 DERIVED METRICS AND FORMULAS

The following metrics are used to evaluate agent performance throughout the appendix.

- **Alignment:** $A = \text{TSR}/\text{CRR}$.
- **Contract gap:** $\Delta_{rc} = \text{CRR} - \text{TSR}$ (in percentage points).

- **Debugging efficiency:** $E = \text{TSR}/\text{ADI}$.
- **Tokens per successful task:** $T^* = \text{Avg. Tokens}/(\text{TSR}/100)$. This measures the average number of tokens consumed to achieve one successful task completion.

The following sections provide the specific numerical data and interpretations corresponding to the visualizations in Figures 5 through 9.

GPT-5 base – DAG-level details

- **DataGovAgent** (TSR 60, CRR 74, ATS 54.91, Avg. 62.97, ADI 3.29, Tokens 34303.72)
  $A = 0.81$; $\Delta_{rc} = 14$; $E = 18.24$; $T^* = 57,173$.
- **ChatDev** (64, 82, 39.67, 61.89, 14.89, Tokens 28607.22)
  $A = 0.78$; $\Delta_{rc} = 18$; $E = 4.30$; $T^* = 44,700$.
- **CAMEL** (32, 74, 16.80, 40.93, 5.00, Tokens 11777.50)
  $A = 0.43$; $\Delta_{rc} = 42$; $E = 6.40$; $T^* = 36,805$.

*Interpretation: On complex DAG-level tasks, DataGovAgent demonstrates the highest debugging efficiency (E=18.24) and strong alignment (A=0.81). However, this comes at the highest token cost per successful task ($T^* = 57,173$). In contrast, CAMEL is the most token-efficient per success ($T = 36,805$) but delivers significantly lower quality (TSR 32, ATS 16.80) and poor alignment. ChatDev offers a middle ground on token efficiency but lags considerably in debugging efficiency.*

GPT-5 base – Operator-level details

- **DataGovAgent** (TSR 64, CRR 88, ATS 55.47, Avg. 69.15, ADI 2.14, Tokens 31503.75)
  $A = 0.73$; $\Delta_{rc} = 24$; $E = 29.91$; $T^* = 49,225$.
- **ChatDev** (43, 69, 33.82, 48.61, 14.47, Tokens 26888.26)
  $A = 0.62$; $\Delta_{rc} = 26$; $E = 2.97$; $T^* = 62,531$.
- **CAMEL** (34, 92, 20.36, 48.79, 4.50, Tokens 9447.75)
  $A = 0.37$; $\Delta_{rc} = 58$; $E = 7.56$; $T^* = 27,788$.

*Interpretation: Even on simpler Op-level tasks, DataGovAgent leads in quality (TSR 64, ATS 55.47) and debugging efficiency (E=29.91). It is also more token-efficient per success than ChatDev ($T = 49,225$ vs. $62,531$). CAMEL remains the most token-efficient overall ($T = 27,788$) but has the worst alignment (A=0.37) and a large correctness gap ($\Delta_{rc} = 58$), indicating that while its raw token usage is low, it struggles to convert runnability into correct solutions.*

Weaker base model (GPT-4o) – token-quality trade-off **DAG-level:**

- **DataGovAgent** (44, 50, 34.52, 42.84, 4.03, Tokens 27192.45): $A = 0.88$; $\Delta_{rc} = 6$; $E = 10.92$; $T^* = 61,801$.
- **ChatDev** (36, 40, 19.12, 31.71, 14.42, Tokens 7261.49): $A = 0.90$; $\Delta_{rc} = 4$; $E = 2.50$; $T^* = 20,171$.
- **CAMEL** (24, 60, 8.47, 30.82, 5.00, Tokens 11925.00): $A = 0.40$; $\Delta_{rc} = 36$; $E = 4.80$; $T^* = 49,688$.

**Operator-level:**

- **DataGovAgent** (63, 89, 52.93, 68.31, 2.12, Tokens 23712.14): $A = 0.71$; $\Delta_{rc} = 26$; $E = 29.72$; $T^* = 37,638$.
- **ChatDev** (43, 63, 34.47, 46.82, 14.20, Tokens 6996.62): $A = 0.68$; $\Delta_{rc} = 20$; $E = 3.03$; $T^* = 16,271$.
- **CAMEL** (29, 91, 14.54, 44.85, 4.40, Tokens 9071.92): $A = 0.32$; $\Delta_{rc} = 62$; $E = 6.59$; $T^* = 31,282$.

*Interpretation: With the weaker GPT-4o model, the trade-offs become more pronounced. DataGov-Agent still achieves the highest quality (TSR/ATS) and debugging efficiency (E), but at a significantly*

*higher token cost per success (T\*). Surprisingly, ChatDev becomes the most token-efficient framework (T\* of 20,171 on DAG and 16,271 on Op), despite its low raw success rate and poor debugging efficiency. This highlights a clear, controllable token-quality frontier where achieving higher quality and development efficiency with DataGovAgent requires a larger token budget.*

### A.7.2 PERFORMANCE VISUALIZATIONS

The following figures provide a comparative visualization of agent performance across different models, task levels, and key metrics.

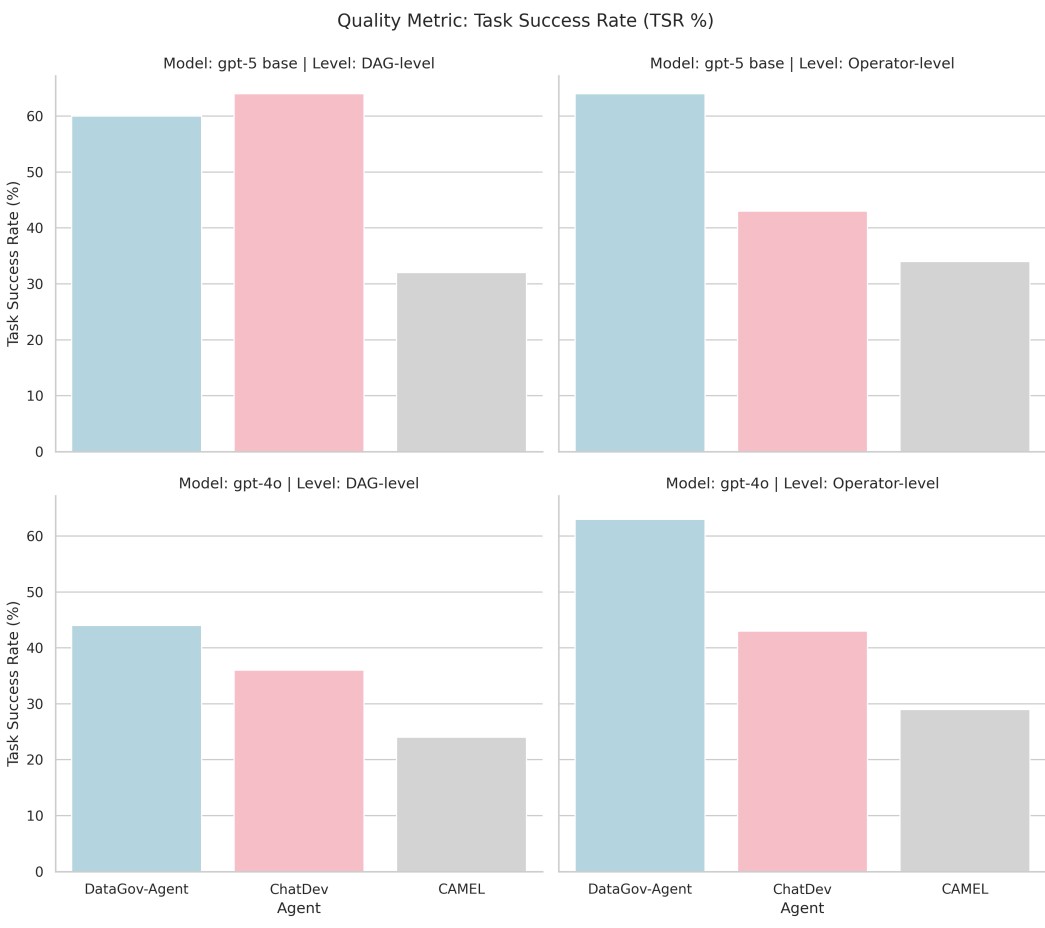

Figure 5: Comparison of Task Success Rate (TSR) across agents, base models, and task levels. TSR measures the percentage of tasks completed successfully.

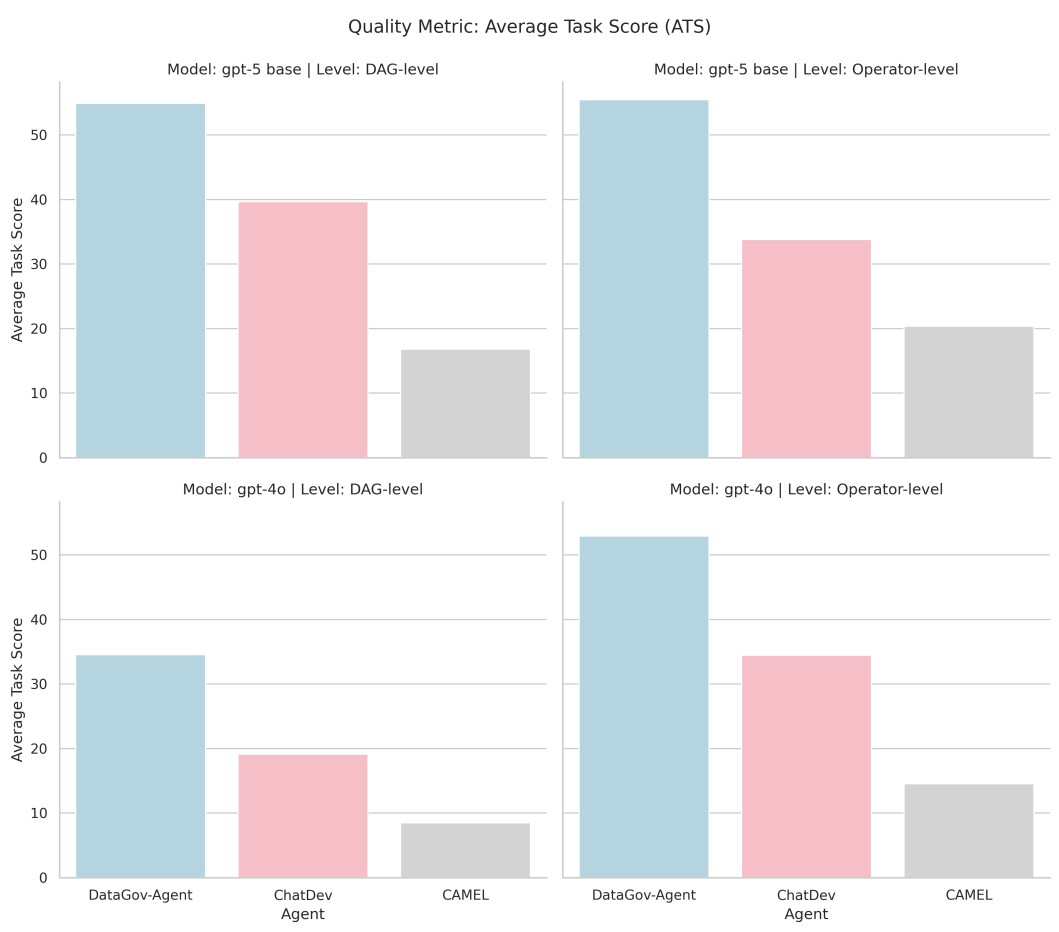

Figure 6: Comparison of Average Task Score (ATS). ATS provides a more nuanced measure of solution quality beyond simple success or failure.

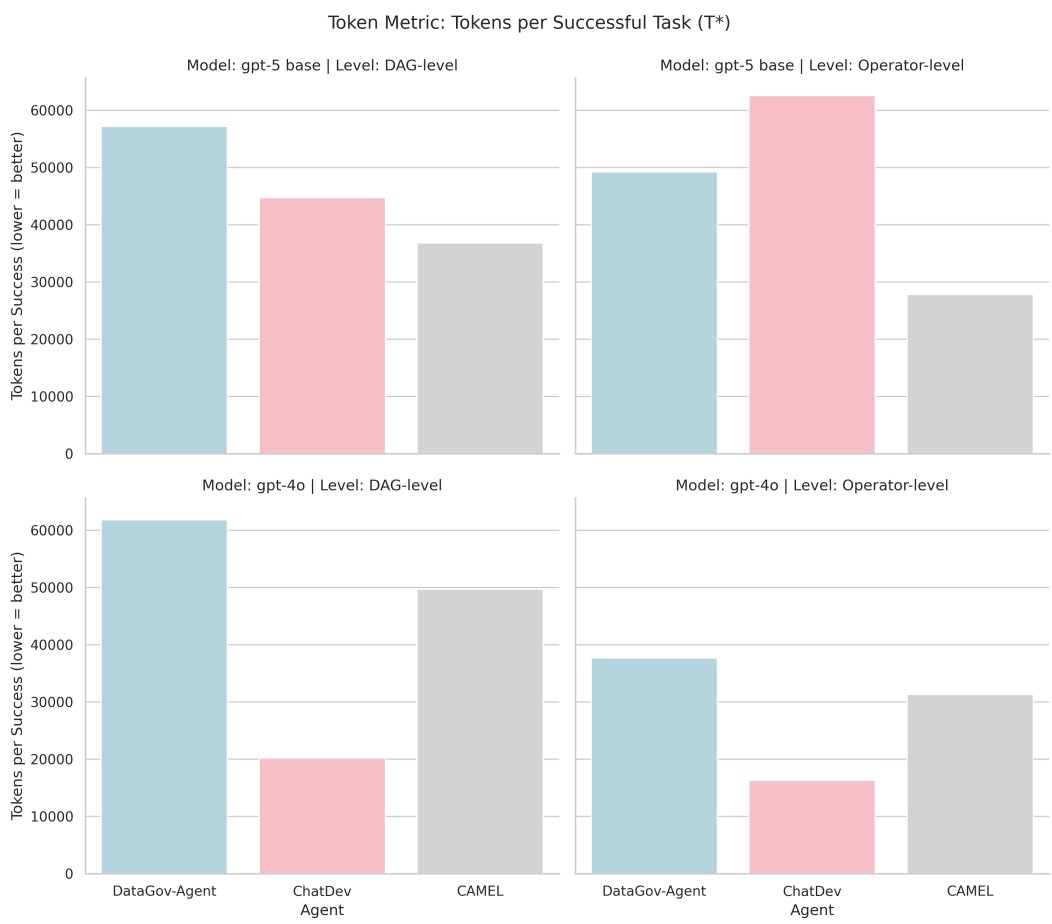

Figure 7: Comparison of Tokens per Successful Task ($T^*$). This metric normalizes average token consumption by the success rate, indicating token-efficiency. Lower values are better.

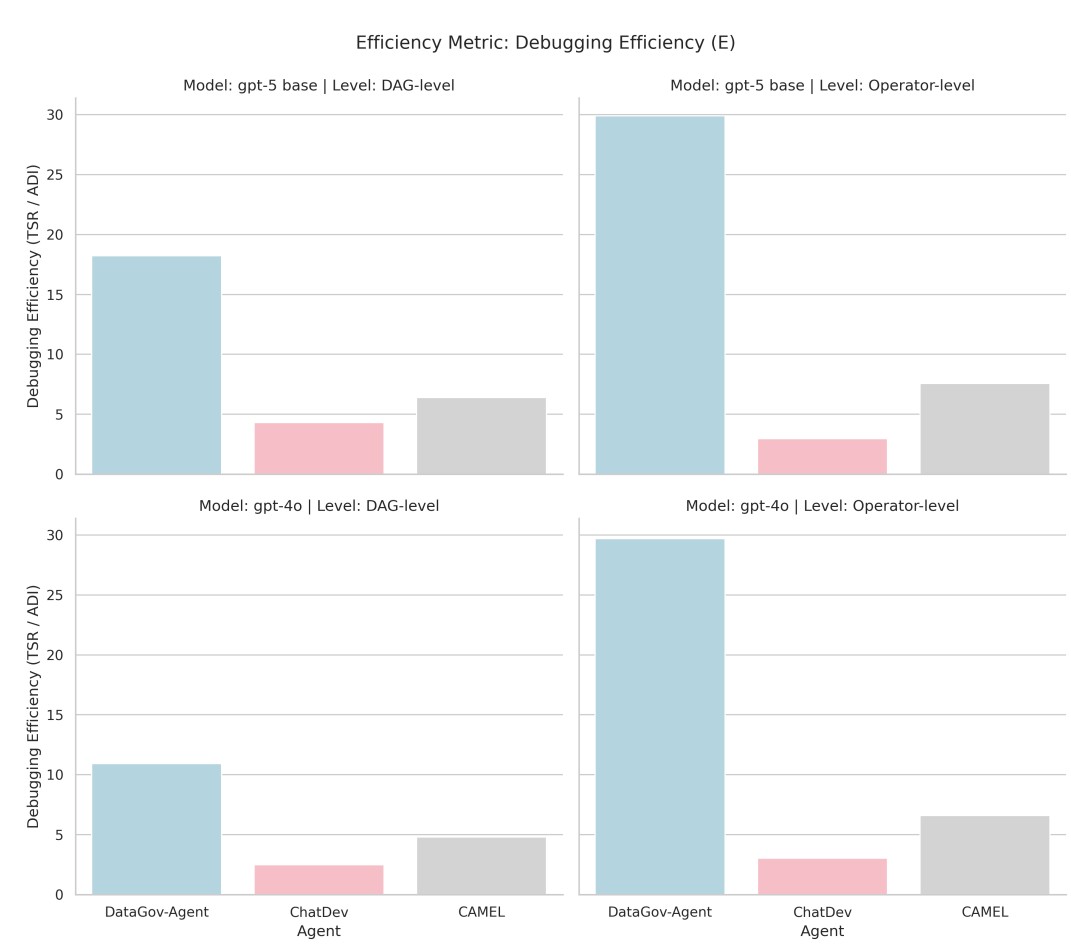

Figure 8: Comparison of Debugging Efficiency ($E$). This metric reflects how many successful tasks are produced per debugging iteration. Higher values are better.

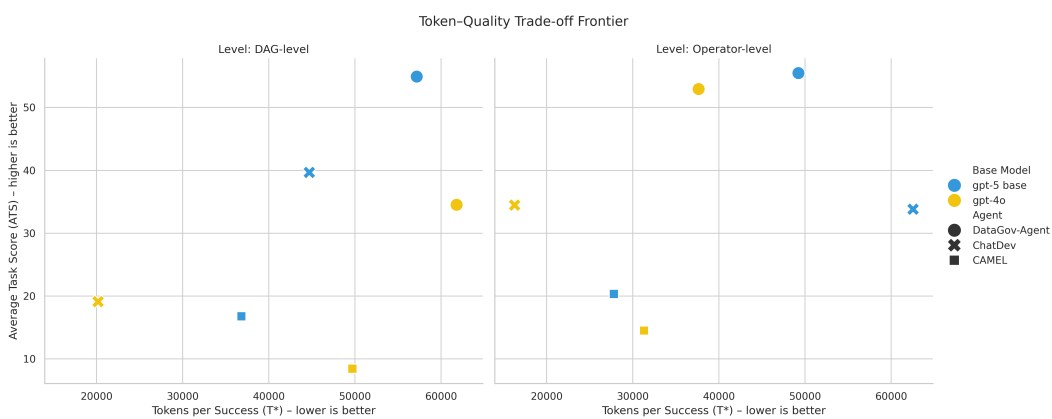

Figure 9: Token–Quality Trade-off Frontier. Relationship between quality (ATS, $y$-axis) and token efficiency ($T^*$, $x$-axis). The ideal position is the top-left corner (high quality, low tokens per success).

### A.7.3 MECHANISM ATTRIBUTION AND ABLATIONS

**Contracts make business correctness executable:** pre-conditions expose type/shape/uniqueness/missing-value assumptions; post-conditions render acceptance criteria as assertions, preventing hidden cross-step assumptions.

**Meta-cognitive feedback turns CRR's blind spots into targeted fixes:** the Evaluator couples failing code spans, stack traces, and violated contracts to produce surgical edits, improving A and reducing $\Delta_{rc}$ with far fewer iterations (higher E). **Ablations (Op-level, GPT-5 base), as shown in Table 8:**

- **w/o Planner:** TSR drops from $64 \rightarrow 38$ ($-26$ pp), CRR $88 \rightarrow 51$, ADI $2.14 \rightarrow 8.75$; ATS $55.47 \rightarrow 31.20$.
- **w/o RAG:** TSR $64 \rightarrow 49$ ($-15$ pp), CRR $88 \rightarrow 65$, ADI $2.14 \rightarrow 5.20$; ATS $55.47 \rightarrow 42.15$.

These confirm that contract-guided planning supplies the right decomposition/ordering, while RAG reduces hallucinations; the Evaluator's meta-cognitive loop converts these into fewer, more effective iterations.

### A.8 ABLATION STUDY

To dissect the contribution of each component within the DATAGOVAGENT framework, we conducted a series of ablation studies on the GovBench Operator-level tasks. We systematically disabled or replaced key modules—the Planner and the RAG mechanism to quantify their impact on overall performance. All experiments were run using GPT-5 as the base model. The results are summarized in Table 8.

Table 8: Ablation study of DataGovAgent on GovBench operator-level tasks. Numbers in brackets show the change () w.r.t. the full model — red = decrease, green = increase.

| Configuration | ATS↑ | TSR↑ | CRR↑ | ADI↓ |
|---|---|---|---|---|
| **DataGovAgent (Full)** | **55.47** | **64.00** | **88.00** | **2.14** |
| *RQ1: Planner's Role* | | | | |
| w/o Planner | 31.20 (-24.27) | 38.00 (-26.00) | 51.00 (-37.00) | 8.75 (+6.61) |
| *RQ2: RAG's Impact* | | | | |
| w/o RAG (Free Generation) | 42.15 (-13.32) | 49.00 (-15.00) | 65.00 (-23.00) | 5.20 (+3.06) |

**RQ1: Is the Planner's high-level DAG planning necessary?** To answer this, we created a variant named **'w/o Planner'**, where the Executor directly receives the raw natural language instruction and attempts to generate the entire solution in one go, bypassing the intent understanding and DAG planning phase. As shown in Table 8, this led to a catastrophic performance drop: the TSR plummeted from 64.00% to 38.00%, and the Average Debug Iterations (ADI) quadrupled. This result strongly indicates that for data governance tasks, which often involve implicit multi-step logic, decomposing the user's intent into a structured, high-level plan is crucial. Without this planning phase, the LLM struggles to manage the complexity, leading to logically flawed or incomplete code that is difficult to debug.

**RQ2: How much does Retrieval-Augmented Generation contribute?** We investigated this by creating the **'w/o RAG'** variant, where the Executor generates code based solely on the abstract operator name provided by the Planner, without retrieving any code examples from the operator library. The performance degradation was significant, with TSR dropping by 15 percentage points. This highlights the value of RAG: grounding the LLM with pre-validated, high-quality code snippets (even if they are not a perfect match) significantly steers it towards generating more correct and robust solutions, reducing hallucinations and logical errors.

## A.9 Details of Human Baseline

To establish a *strong* human baseline for DATAGOVAGENT, we evaluated the performance of experienced data-science practitioners on a subset of GOVBENCH. We recruited five data-science experts, each with more than five years of professional experience in data engineering and analysis.

To ensure a fair comparison, the experts were **granted unrestricted access to the same GPT-5 model** through an interactive chat interface. They could issue any number of queries but still had to manually synthesize, test, and iterate on a final Python script. Each expert completed ten randomly sampled tasks—five Operator-level and five DAG-level. We measured both the TSR and the average wall-clock time from start to finish.

Table 9 presents the full results of this comparison.

Table 9: Performance of Human Experts vs. DataGovAgent on GovBench Subset.

| Method | Task | TSR↑ | Avg. Time (min)↓ |
|---|---|---|---|
| Human Experts + GPT-5 | Op | 84.00 | 14.2 |
| Human Experts + GPT-5 | DAG | 25.00 | 24.5 |
| DataGovAgent (GPT-5) | Op | 64.00 | 3.5 |
| DataGovAgent (GPT-5) | DAG | 60.00 | 4.7 |

Our study reveals complementary strengths rather than uniform dominance: the agent excels on complex DAG-level tasks, whereas humans achieve higher accuracy on operator-level tasks. Concretely:

- **Operator-level tasks.** While human experts achieved a higher TSR, DATAGOVAGENT was approximately **4.1× faster** on average (3.5 min vs. 14.2 min).

- **DAG-level tasks.** For more complex tasks, the agent's advantage was twofold: it achieved an accuracy **35 percentage points higher** than the experts and reduced completion time by roughly **81%** (4.7 min vs. 24.5 min).

These findings suggest that, for well-specified data-governance workloads, a fully automated LLM-centric agent can translate the reasoning and coding capabilities of GPT-5 into effective end-to-end execution, particularly on complex DAG workflows. Human expertise remains crucial for open-ended problem formulation, strategic oversight, and final validation; our results show that routine to moderately complex data-processing tasks can be accomplished substantially faster by the agent, with higher accuracy on DAG-level tasks and lower accuracy on operator-level tasks relative to humans.

A.10 METRICS

Table 10: Evaluation Metrics for GovBench

| Metric | Abbr. | Calculation | Description |
|--------|-------|-------------|-------------|
| Average Task Score | ATS | $\frac{100}{N_t} \sum_{i=1}^{N_t} S_i$ | Represents the ATS across all tasks, reflecting the overall quality of the generated solutions. A higher ATS indicates better overall performance. |
| Task Success Rate | TSR | $\frac{N_{\text{succ}}}{N_t}$ | The proportion of tasks that fully achieve the "business objective." A task is deemed successful if its evaluation score is 1.0. This is the core metric for measuring task completion quality. |
| Code Runnable Rate | CRR | $\frac{N_{\text{run}}}{N_{\text{gen}}}$ | The proportion of generated code scripts that can be executed directly without any uncaught errors. This measures the basic usability of the code. |
| Avg. Score | – | $S_{avg}$ | The average value of the ATS, TSR, and CRR metrics. This metric provides an overall score by averaging these three indicators. |
| Average Debug Iterations | ADI | $\frac{1}{N_t} \sum_{i=1}^{N_t} D_i$ | The average number of "generate → execute → evaluate" cycles required for a task to succeed. This measures the debugging efficiency of the agent framework. |
| Avg. Tokens | – | $T_{avg}$ | The average number of tokens consumed to complete each individual task. This metric evaluates the token efficiency for every single task. |
| Total Cost | – | $C_i$ | The monetary cost required to complete each individual task, calculated based on openai LLM API pricing. This metric evaluates the economic efficiency for every single task. |
| Generation Time | – | $T_{\text{gen}}$ | Total wall-clock time (in seconds) consumed by the LLM to generate all task code solutions. This reflects the raw code synthesis efficiency. |
| Execution Time | – | $T_{\text{exec}}$ | Total wall-clock time (in seconds) consumed by running all generated task code solutions. This reflects the runtime efficiency of the produced code. |

**Where:** $N_t$ is the total number of tasks; $S_i$ is the evaluation score for task $i$; $N_{\text{succ}}$ is the number of successful tasks; $N_{\text{run}}$ is the number of runnable scripts; $N_{\text{gen}}$ is the total number of generated scripts; $D_i$ is the number of debug iterations for task $i$; $C_i$ is the monetary cost for each individual task; $T_{\text{gen}}$ is the total generation time across all tasks; and $T_{\text{exec}}$ is the total execution time across all tasks. **Notes:** ATS = 100 × mean per-task score (each task score $\in [0,1]$). TSR/CRR are proportions reported as percentages. Higher is better unless noted.

### A.11 PROMPTS

Here's some prompt templates used in Benchmark Building.

Prompt 1: Prompt for building DAG tasks.

```
### Task Description
You are given a sequence of task descriptions. Each task description
    defines a part of a complex task or operation. The task descriptions
    are part of a larger, multi-step process that will form a
    comprehensive, integrated task. Your objective is to generate a new,
    high-level task objective that combines the individual task
    descriptions into a coherent and complex task. This task must
    challenge the model's ability to handle intricate data governance
    problems.

### Instructions
1. Combine the given task descriptions into a single, cohesive task that
    requires handling multiple steps.
2. Incorporate multiple aspects of the given task descriptions into the
    final task description to present a significant challenge to data
    governance.

### Task Descriptions
- {task_1}
- {task_2}
- {task_3}
- ...

### Generated Comprehensive Task
{generated_task}
```

Prompt 2: Prompt for reverse prompt.

```
### Original Task Objective
You are given the following task objective. Your goal is to achieve the
    stated objective using the provided data examples.

### Task Description
{original_task_description}

### Reversed Task Objective
Now, your task is to generate a reversed task objective based on the
    provided task description. The reversed objective should shift the
    focus from achieving the task goal to intentionally introducing noise
     into the data. Instead of performing actions such as classification,
     imputation, or any other task goal, the goal is to create challenges
     or distortions in the data. For example, if the original task
    involves classification, the reversed task should focus on
    introducing noise such as mislabeling or irrelevant features in the
    data.

### Data Examples
Here are the provided data examples related to the original task:

- {example_1}
- {example_2}
- {example_3}
- ...

### Generated Reversed Task Objective
{generated_reversed_task}
```

Prompt 3: Prompt for noisy injection.

```
### Reversed Task Objective
You are given the following reversed task objective. This objective
    describes how to intentionally introduce noise into the dataset.

{reversed_task_objective}

### Data Examples
Here are some sample data records that illustrate the structure and
    format of the dataset:

- {example_1}
- {example_2}
- {example_3}
- ...

### Instruction
Write executable Python code that introduces the noise into the dataset
    as described in the reversed task objective.
The code should:
1. Take as input a dataset file (format consistent with the given
    examples).
2. Implement the noise generation specified in the reversed task
    objective.
3. Output the modified dataset to required file path in the same format
    as the input.
4. Ensure reproducibility (e.g., by setting a random seed if randomness
    is used).

### Expected Output
Provide only the Python code that implements the noise injection process.
The code must be complete and runnable.
```

Prompt 4: Prompt for evaluation scripts generation.

```
### Task Description
You are given a data governance task description:

{task_description}

### Data Samples
Here are some representative ground truth (expected) data samples:

{gt_samples}

Here are some representative processed data samples:

{processed_samples}

### Instruction
Write a Python evaluation script that compares the processed dataset
    against the ground truth dataset and outputs a quantitative score
    between 0 and 1, reflecting the m o d e l s effectiveness in completing
     the task.

The evaluation should:
1. Load the ground truth and processed datasets from file paths provided
    as arguments.
2. Use evaluation metrics appropriate for the task category:
    - Filtering: F1 Score (balance of precision and recall in filtering
     unwanted entries).
    - Refinement: Accuracy (correctness of standardized or transformed
     data fields).
    - Imputation: Completion Rate / Imputation Accuracy (ability to
     correctly fill in missing values).
```

```
   - Deduplication & Consistency: Duplicate Reduction Rate or Consistency
     Score (removal of duplicates or ensuring consistent values).
   - Data Integration: Integration Accuracy (accuracy of merging
    heterogeneous datasets, resolving conflicts).
   - Classification & Labeling: Accuracy, Precision, Recall, F1 Score (
    standard classification metrics).
3. Output the evaluation result as a dictionary with the key `"eval_score
    "` and the corresponding score (float between 0 and 1).
4. Print the dictionary as the final output.

### Expected Output
Provide only the Python code for the evaluation script.
The code should be complete and runnable, following this template
    structure:

```python
def evaluate(processed_path):
    expected_path = get_gt()
    expected = load_gt(expected_path)
    processed = load_processed(processed_path)

    # implement task-specific evaluation logic here ...

    result = {"eval_score": <score>}
    print(result)
```

To enhance reproducibility and review transparency, this appendix discloses several prompts used in our experiments (including intent identification, pipeline assembly, operator retrieval, and code debugging). We emphasize that these prompts only support a subset of "minimum viable" functionality and are not sufficient on their own to constitute the full contract-driven Planner–Executor–Evaluator framework described in the main paper.

Prompt 5 present the detailed prompts for Planner.

Prompt 5: Prompt for Intent Understanding.

```
[Role] You are an intent analysis robot. You need to identify the user's
    explicit intent from the conversation and analyze the user's data
    processing requirements based on the conversation content.
[Task]

You need to determine whether the user's current requirement is for a
    single operator or a complete pipeline, and set is_single_operator (
    true only if a single operator is required, otherwise false) and
    is_pipeline (true if pipeline processing is required, otherwise false
    ) accordingly.
You need to summarize the user's processing requirements in detail based
    on the conversation history, and always provide a natural language
    response as the value of assistant_reply.
[Input Content] Conversation history: {history} Current user request: {
    target}
[Output Rules]
Reply only in the specified JSON format.
Do not output anything except JSON.
[Example]
{
"is_single_operator": false,
"is_pipeline": true,
"assistant_reply": "I will recommend a suitable data processing pipeline
    based on your needs.",
"reason": "The user explicitly requested a recommendation, wants to
    process data related to mathematics, and hopes to generate pseudo-
    answers.",
```

```
"purpose": "According to the conversation history, the user does not need
    a deduplication operator, hopes to generate pseudo-answers, and
    wants to keep the number of operators at 3."
}
```

Prompt 6 Prompt for the agent in recommend Module.

Prompt 6: Prompt for the agent in recommend Module.

```
[ROLE]
You are a data governance workflow recommendation system. Based on the
    provided context, automatically select the appropriate operator nodes
    and assemble them into a complete data processing pipeline.

[INPUT]
You will receive the following information:
- Workflow requirements to be satisfied:
  {workflow_bg}
- Sample data information:
  {local_tool_for_sample}
- List of available operators:
  {operators}

[OUTPUT RULES]
1. Select suitable operator nodes from the available operators and
    assemble them into a complete processing pipeline. Output in the
    following JSON format:
    {"edges":[{"source":node0,"target":node1},{"source":node1,"target":
    node2}]}
2. Provide your reasoning for the selection in the following JSON format:
    {"reason": "Please explain your reasoning in detail here. For example:
     The pipeline includes multi-level data preprocessing and quality
    filtering, performing language filtering, format standardization,
    noise removal, privacy protection, length and structure optimization,
     and symbol and special character handling sequentially to ensure the
     text content is standardized, rich, and compliant."}
3. Verify that the constructed pipeline satisfies all requirements,
    especially {workflow_bg}.
4. Check the edges field to ensure all nodes are valid node fields from
    the available operators.
5. For each operator, specify the conditions under which it can continue
    execution, using the following format:
    "node1": {
       "Score": { "operator": ">", "value": 0.5 }
    }.
```

Prompt 7 Prompt for the agent in op lib Module.

Prompt 7: Prompt for the agent in op lib Module.

```
[ROLE]
You are an expert in data operator retrieval.

[TASK]
Based on the provided operator content {get_operator_content}, user
    requirement {target}, and operator names {op_name}, identify the top
    {top-k} most similar operator names from the operator library and
    provide your reasoning.

[INPUT FORMAT]
The input includes:
- Operator content (get_operator_content)
- User requirement (target)
- Operator names (op_name)
```

```
[OUTPUT RULES]
1. Strictly return the content in the JSON structure shown below. Do not
    include any extra content, comments, or additional fields.
2. You must return exactly {top-k} operator names in all cases.

JSON output example:
{
  "match_operators": [
    "OperatorName1",
    "OperatorName2",
    "OperatorName3",
    "OperatorName4"
  ],
  "reason": "xxx"
}
```

Prompt 8 Prompt for the agent in write op Module.

Prompt 8: Prompt for the agent in write op Module.

```
[ROLE]
You are an expert in data operator development.

[TASK]
Refer to the example operator {example} and write a new operator based on
    the requirements described in {target}.

[INPUT FORMAT]
Input includes:
- Example operator (example)
- Target description (target)

[OUTPUT FORMAT]
Please output in the following JSON structure:
{
  "code": "Complete source code of the operator",
  "desc": "Brief description of the operators function and its input/
    output"
}

[RULES]
1. Carefully analyze and understand the structure and coding style of the
    example operator.
2. Write operator code that fully meets the functional requirements of {
    target} and can run independently. Do not include any extra code or
    comments.
3. Only output the two fields 'code' (the complete operator code as a
    string) and 'desc' (a concise explanation of the operators
    function and its input/output), strictly following the JSON format.
4. If the operator requires using an LLM, the __init__ method must
    include the llm_serving field.
5. All output files generated by the operator must be in the same
    directory as the current file (os.path.dirname(__file__)).
```

Prompt 9 Prompt for the agent in debug Module.

Prompt 9: Prompt for the agent in debug Module.

```
[ROLE]
You are an expert in code debugging and correction.

[TASK]
Given the original code, error message, requirement, JSON data fields,
    and reference code, minimally modify the original code to fix the
    error. Ensure your corrections are precise and focus on issues such
```

```
    as key alignment or import errors. Output the corrected code and your
     reason for modification strictly in JSON format, and follow all
    specified requirements.

[INPUT]
You will receive the following information:
- The original code: {code}
- The error message: {error}
- The requirement: {target}
- The JSON data fields processed in the target code: {data_keys}
- Reference code retrieved: {cls_detail_code}

[OUTPUT RULES]
1. Strictly return your response in JSON format, including: the complete
    corrected code, your reason for the modification, and any additional
    files that may be needed to better resolve the error. For example: {"
    code": xxx, "reason": xxx}
2. Ensure that the operator output file is in the same directory as the
    currently executing file (os.path.dirname(file)).
3. Do not include any extra keys, explanations, comments, or markdown
    syntax.
4. The returned code must include the if __name__ == '__main__': block,
    so that the file can be run independently.
5. The output must be in JSON format!!!!
6. You must use the files specified in <INPUT_FILES>{INPUT_FILES}</
    INPUT_FILES> as input.
```

## A.12 LLM USAGE STATEMENT

In this research, large language models (LLMs) were used to assist in certain stages, as detailed below:

1. During the writing process, GPT-5 was utilized for language polishing and grammar correction.

2. LLMs were used to assist in code generation and the development of visualization scripts.

3. All research ideas, experimental designs, data analyses, and conclusions were independently conceived and determined **by the authors.**

