# OpenReview forum: "GovBench: From Natural Language to Executable Pipelines, A New Benchmark for Data Governance Automation"
_ICLR.cc/2026/Conference — ICLR 2026 Conference Withdrawn Submission_

### Official Review · Reviewer_NoRH · 2025-10-29

**Soundness:** 2
**Presentation:** 3
**Contribution:** 2
**Rating:** 2
**Confidence:** 3

**Summary:**

This paper addresses two key gaps in automated data governance. First, it identifies the lack of a realistic, standardized benchmark for evaluating end-to-end governance workflows and introduces GovBench, the first hierarchical benchmark tailored for this purpose. Second, it shows that existing models struggle with complex, multi-step governance tasks due to weak planning and error correction. To address this, the authors propose DataGovAgent, an end-to-end agent framework with a Planner–Executor–Evaluator architecture: the Planner extracts user intent and formal contracts, the Executor uses retrieval-augmented generation from a curated operator library, and the Evaluator performs contract-based debugging. Extensive experiments show (1) benchmark performance of existing LLMs, (2) strong improvements of DataGovAgent over baselines like ChatDev and CAMEL, and (3) superior performance over human experts on complex DAG tasks.

**Strengths:**

This paper makes a contribution to the field of automated data governance by introducing GovBench, the first benchmark for data governance automation tailored to evaluate LLMs on end-to-end data governance tasks with task-specific metrics and diverse pipeline structures. The authors conduct a thorough and systematic benchmarking of both open- and closed-source LLMs, providing actionable insights for practitioners interested in deploying these models for real-world data science workflows. Furthermore, the proposed DataGovAgent demonstrates impressive gains over strong agent baselines like ChatDev and CAMEL on the proposed benchmark tasks. These results underscore the practical potential of automated governance pipeline construction for real-world data science applications.

**Weaknesses:**

1. The significance of this benchmark/work is unclear, despite the authors’ claim that the benchmark is "realistic" (see Question 1 and 3 for more details).
2. The soundness about the DataGovAgent vs. human experts remains questionable (see Question 2 below for more details)
3. The rationale and motivation behind (1) the construction process of the GovBench benchmark and (2) the design choices of the DataGovAgent remain unclear (see Question 4 below for more details).

**Questions:**

1. Why is this benchmark necessary? Specifically, what problems does it reveal that existing benchmarks fail to capture—either quantitatively or qualitatively? How significant are these problems in real-world applications? While Section A.1 provides a comparison, it does not clearly articulate what is lacking in prior benchmarks or justify the real-world importance of addressing those gaps.
2. On the DataGovAgent vs humans experiments: Can the authors clarify how the 25% task success rate (TSR) was measured? Maybe I missed some details, I am confused as, for 5 experts each solve 5 DAG tasks (25 tasks in total), achieving 25% task success rate (TSR$:=N_{succ}/N_t$) means solving $N_{succ}=6.25$ tasks successfully, how?
3. On the DataGovAgent vs humans experiments: Any insights why humans experts are so poorly-performed? Essentially how would the authors justify the realism and value of the work/benchmark?
4. The paper clearly outlines the benchmark creation process, but the motivation behind each step remains unclear. What are the technical challenges each operations intended to address? I.e., what are the technical challenges/problems each operation was desgined to solve? I encourage the authors to clarify the rationale behind each design choice.

---

### Official Review · Reviewer_uuCg · 2025-10-31

**Soundness:** 2
**Presentation:** 2
**Contribution:** 2
**Rating:** 2
**Confidence:** 2

**Summary:**

This paper introduces GovBench, a hierarchical benchmark for data governance automation. It also proposes DataGovAgent, an multi-agent workflow with a Planner-Executor-Evaluator architecture for translating natural language into verified governance pipelines. Experiments on GovBench show DataGovAgent outperforms SOTA models and general agent frameworks, boosting complex DAG-task ATS.

**Strengths:**

1. GovBench overcomes the limitations of existing snippet-focused benchmarks. The proposed hierarchical tasks (operator/DAG-level) and targeted noise injection simulate real-world data governance scenarios.
2. DataGovAgent’s architecture is intuitive. Contract-guided planning, retrieval-augmented generation, and meta-cognitive debugging directly solve complex workflow decomposition and error-correction issues.
3. Comprehensive experiments (vs. SOTA models, agent frameworks, humans) clearly demonstrate the benchmark’s challenge and the framework’s effectiveness.

**Weaknesses:**

1. Motivation and workflow design lack theoretical rigor, leaning overly toward engineering. The paper frames its work primarily around "fixing practical tool limitations" but fails to anchor this in broader research gaps. While DAG construction (via LCS-aware algorithms) and noise injection (reverse-objective method) are technically detailed, the paper offers little analysis of their theoretical significance (e.g., why these designs effectively test model capabilities). The result is work that reads more like a tool-building project than a research contribution that advances conceptual understanding of data governance automation.
2. Limited novel research insights. Core components like "contract-guided planning" and "meta-cognitive debugging" are presented as implementation steps (e.g., "how to extract contracts" or "how to generate debug feedback"). The paper doesn’t elaborate on why these methods outperform alternatives beyond experimental results.

**Questions:**

See weakness. Overall, core components are presented as implementation steps, lacking insightful analysis, making the work read more like a tool-building project than a research paper.

---

### Official Review · Reviewer_VPeG · 2025-11-01

**Soundness:** 2
**Presentation:** 1
**Contribution:** 3
**Rating:** 2
**Confidence:** 3

**Summary:**

The authors propose a new data governance benchmark to improve automation of data governance. They show that their benchmark improves performance quite a bit (though at a substantial token cost).

**Strengths:**

- I'm always happy to see more benchmarks, especially on a topic like this that has not received enough attention so far. The cited study by Ahmadi et al. really does motivate it exceptionally well.
- The evaluation on the framework is quite thorough, with each aspect of it quite well thought out (multiple evaluation metrics, etc.).

**Weaknesses:**

- More than anything else, I think this paper needs quite a bit of a rewrite (or at least a lot of surgical alterations). The level of the writing is unnecessarily complicated and makes understanding this work exceptionally hard (even for an expert and a native English speaker). It feels almost to me that the authors are trying to leverage some of the Dr. Fox findings in their work. Some examples (taken from the introduction, though many more exist): "schema drift", "synthesize", "evaluation gap", and "meta-cognative". For a paper about benchmarking, this is a big problem. Maybe an even bigger problem is that there's no background section to ease this at all? There's a reason we have them. A benchmarking paper should be one of the most accessible kinds of papers. I'd want to see a fair bit of edits here before I think this paper can actually have any practical use to the community. I understand that this can be hard to address, so I'm open to being less strict on this if this is the only holdout point amongst all the reviews.
- I need more details on the "data science experts." Are these the authors? Are they friends of the authors? Are they random people found on the street? Were they compensated? There are generally ethical procedures required while experimenting with human subjects. Were these followed?
- There are some minor grammatical things that might necessitate a grammar check. For example: the citation at line 171 should be a citet-style, the mention of the CSV to JSONL conversion in line 175 should be removed (not relevant), there's a space missing around the ampersand in line 182, and the citations on line 198 make no sense there.

**Questions:**

See Weaknesses.

---

### Official Review · Reviewer_WGB3 · 2025-11-03

**Soundness:** 3
**Presentation:** 2
**Contribution:** 2
**Rating:** 4
**Confidence:** 4

**Summary:**

The paper introduces a new benchmark for data governance automation: GovBench featuring 150 tasks designed to evaluate the ability of AI models to automate data science workflows. It includes both operator (i.e., filtering, refinement, imputation, de-duplication, integration and labelling) and complex, Directed Acyclic Graph-level (DAG-level) tasks. They introduce and test on the resulting benchmark a new framework for end-to-end data governance: DataGovAgent, based on a planning, execution and evaluation phases with retrieval-augmented code generation and iterative structured-feedback debugging.

**Strengths:**

- DataGovAgent outperforms existing state-of-the-art LLMs and agents, demonstrating better correctness, debugging efficiency, and task success rates.
- Extensive results and comparisons against strong baselines including both open- and closed-source LLMs and existing agentic frameworks.
- The benchmark has been open-sourced, thereby facilitating future research on agentic systems for data governance.

**Weaknesses:**

- While the noise injection approach enhances realism, the paper lacks a thorough discussion on how the benchmark results translate to actual operational data governance scenarios.
- Dependence on Appendix: The main body of the manuscript frequently refers to material presented in the Appendix, such as Figures 4 and Tables 9 and 10. I recommend moving these items into the main text or, alternatively, limiting the related discussions to the Appendix to ensure that the main body of the paper is more self-contained.
- The rationale behind generating a prompt that generates code for evaluating predicted outputs could be more explicitly motivated and discussed in the manuscript.
    - Including a discussion about how variation in the generated evaluation code is controlled, or providing evidence of reproducibility of evaluation outputs across several runs should help.
- How does GovBench differentiate itself from other multi-step planning datasets, such as GAIA (Mialon et al. (2023)

Grégoire Mialon, Clémentine Fourrier, Craig Swift, Thomas Wolf, Yann LeCun, and Thomas Scialom. GAIA: a benchmark for General AI Assistants. Preprint, arXiv:2311.12983.

**Questions:**

- Does the noise injection involve only changes to the data from the original 30 tables sourced from Statista (2025)?
- The manuscript does not provide details on the number of data scientists involved in dataset curation and task creation.
- It is not fully clear how the `original_task_description` input in Prompt 2 is generated or obtained during the noise introduction stage. Is it coming from data scientists that participated in the dataset curation process?
- The `README` file in the provided repository could be improved by including further instructions on how to navigate the codebase and utilise the included data.

---

### Note · Authors · 2025-11-12

**Comment:**

We would like to withdraw our submission due to ongoing revisions and further experiments. We appreciate the reviewers' efforts and hope to resubmit an improved version in the future. We are also among the earliest to prepare for the next conference.

**Withdrawal Confirmation:**

I have read and agree with the venue's withdrawal policy on behalf of myself and my co-authors.